# Reparameterization invariance in approximate Bayesian inference

**Hrittik Roy†, Marco Miani†**
Technical University of Denmark
`{hroy, mmia}@dtu.dk`

**Carl Henrik Ek**
University of Cambridge,
Karolinska Institutet
`che29@cam.ac.uk`

**Philipp Hennig, Marvin Pförtner, Lukas Tatzel**
University of Tübingen, Tübingen AI Center
`{philipp.hennig, lukas.tatzel,`
`marvin.pfoertner}@uni-tuebingen.de`

**Søren Hauberg**
Technical University of Denmark
`sohau@dtu.dk`

## Abstract

Current approximate posteriors in Bayesian neural networks (BNNs) exhibit a crucial limitation: they fail to maintain invariance under reparameterization, i.e. BNNs assign different posterior densities to different parametrizations of identical functions. This creates a fundamental flaw in the application of Bayesian principles as it breaks the correspondence between uncertainty over the parameters with uncertainty over the parametrized function. In this paper, we investigate this issue in the context of the increasingly popular linearized Laplace approximation. Specifically, it has been observed that linearized predictives alleviate the common underfitting problems of the Laplace approximation. We develop a new geometric view of reparametrizations from which we explain the success of linearization. Moreover, we demonstrate that these reparameterization invariance properties can be extended to the original neural network predictive using a Riemannian diffusion process giving a straightforward algorithm for approximate posterior sampling, which empirically improves posterior fit.

## 1 Introduction

Bayesian deep learning has not seen the same degree of success as deep learning in general. Theoretically, Bayesian posteriors *should* be superior to point estimates (Devroye et al., 1996), but the practical benefits of having the posterior are all too often not significant enough to justify their additional computational burden. This has raised the question if we even *should* attempt to estimate full posteriors of all network parameters (Sharma et al., 2023).

As an example, consider the Laplace approximation (MacKay, 1992), which places a Gaussian in weight space through a second-order Taylor expansion of the log-posterior. When applied to neural networks, this is known to significantly *underfit* and assign significant probability mass to functions that fail to fit the training data (Lawrence, 2001; Immer et al., 2021a). Fig. 2 (top-left) exemplifies this failure mode for a small regression problem. Interestingly, this behavior is rarely observed outside neural network models, and the failure appears linked to Bayesian deep learning.

Recently, the *linearized Laplace approximation (LLA)* has been shown to significantly improve on Laplace's approximation through an additional linearization of the neural network (Immer et al., 2021b; Khan et al., 2019). We are unaware of any theoretical justification for this rather counterintuitive result: *why would an additional degree of approximation improve the posterior fit?*

---

†Equal contribution authors listed in random order.

38th Conference on Neural Information Processing Systems (NeurIPS 2024).

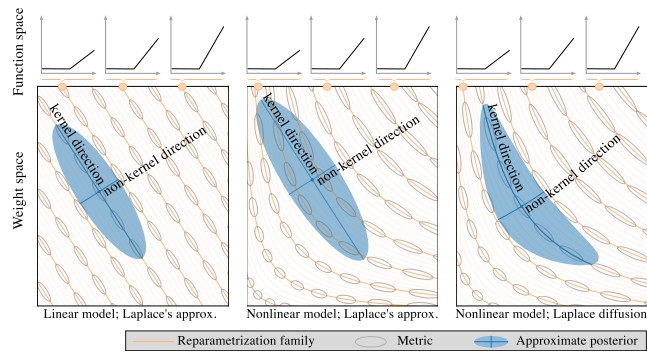

Figure 1: The *weight space* of a neural network (Eq. 1) overparametrizes the associated *function space*. This induces families (orange) of weights corresponding to the same functions. Model linearization (left) linearizes these families. In nonlinear models, Gaussian weight distributions (center) do not adapt to the families, while our geometric diffusion (right) captures the associated invariance with a metric (gray ellipses).

We will show that the failures of Bayesian deep learning can partly be explained by insufficient handling of *reparameterizations* of network weights, while the LLA achieves infinitesimal invariance to reparameterizations. To motivate, consider the simple network (Fig. 1)

$$f(x) = w_1 \text{ReLU}(w_2 x); \qquad f : \mathbb{R} \to \mathbb{R}. \tag{1}$$

This can be *reparametrized* to form the same function realization from different weights as $f(x) = w_1/\alpha \, \text{ReLU}(\alpha w_2 x)$ for any $\alpha > 0$. That is, the weight-pairs $(w_1, w_2)$ and $(w_1/\alpha, \alpha w_2)$ correspond to the same function even if the weights are different (Fig. 1, center).

Thus the approximate posterior cannot reflect the fundamental property of the true posterior, that it should assign a single unique density to a function regardless of its parametrization.

**In this paper**, we analyze the reparameterization group driving deep learning and show that it is a pseudo-Riemannian manifold, with the *generalized Gauss-Newton (*GGN*)* matrix as its pseudo-metric. We prove that the commonly observed underfitting of the Laplace approximation (Fig. 2, top left) is caused by high in-distribution uncertainty in directions of reparameterizations (Fig. 2, top center).

We develop a reparametrization invariant diffusion posterior that proveably does not underfit despite using the neural network predictive (Fig. 2, center row). Figure 1 (right) visualizes how this posterior adapts to the geometry of

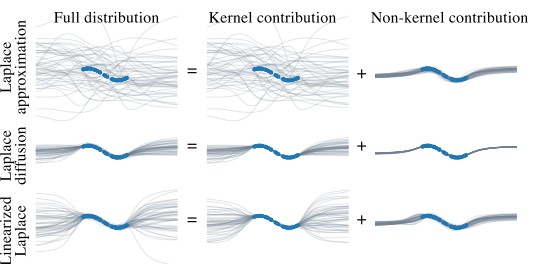

Figure 2: The *function space* is decomposed into directions of *reparameterizations* (kernel) and *functional change* (non-kernel). We improve the posterior fit by concentrating probability mass on directions of functional change.

reparameterizations, thereby not underfitting. The diffusion can be simulated with a multi-step Euler-Maruyama scheme from which the linearized Laplace approximation (LLA) is a single-step. This link implies that the LLA *infinitesimally* is invariant to reparameterizations, due to the otherwise counterintuitive linearization (Fig. 1, left). Experimentally, our diffusion consistently improves posterior fit, suggesting that reparameterizations should be given more attention in Bayesian deep learning.

## 2 Background: Laplace approximations

Let $f_{\mathbf{w}} : \mathbb{R}^I \to \mathbb{R}^O$ denote a neural network with weights $\mathbf{w}$, and define a likelihood $p(\mathbf{y}|f_{\mathbf{w}}(\mathbf{x}))$ and a prior $p(\mathbf{w})$. *Laplace's approximation* (MacKay, 1992) performs a second-order Taylor expansion of the log-posterior around a mode $\hat{\mathbf{w}}$. This results in a Gaussian approximate posterior $\mathcal{N}(\mathbf{w}|\hat{\mathbf{w}}, -\mathbf{H}_{\hat{\mathbf{w}}}^{-1})$, where $\mathbf{H}_{\mathbf{w}}$ is the Hessian matrix. *The linearized Laplace approximation* (Immer et al., 2021b; Khan et al., 2019) further linearize $f_{\mathbf{w}}$ at a chosen weight $\hat{\mathbf{w}}$, i.e. $f_{\mathbf{w}}(\mathbf{x}) \approx f_{\text{lin}}^{\hat{\mathbf{w}}}(\mathbf{w}, \mathbf{x})) = f_{\hat{\mathbf{w}}}(\mathbf{x}) + \mathbf{J}_{\hat{\mathbf{w}}}(\mathbf{x})(\mathbf{w} - \hat{\mathbf{w}})$, where $\mathbf{J}_{\hat{\mathbf{w}}}(\mathbf{x}) = \partial_{\mathbf{w}} f_{\mathbf{w}}(\mathbf{x})|_{\mathbf{w}=\hat{\mathbf{w}}} \in \mathbb{R}^{O \times D}$ is the Jacobian of $f_{\mathbf{w}}$. Here $D = \dim(\mathbf{w})$ denotes the number of parameters in the network. Applying the usual Laplace approximation to the linearized model yields an approximate posterior (Immer et al., 2021b),

$$q(\mathbf{w}|\mathcal{D}) = \mathcal{N}\left(\mathbf{w} \mid \hat{\mathbf{w}}, (\text{GGN}_{\hat{\mathbf{w}}} + \alpha \mathbf{I})^{-1}\right) \qquad \text{GGN}_{\hat{\mathbf{w}}} = \sum_{n=1}^{N} \mathbf{J}_{\hat{\mathbf{w}}}(\mathbf{x}_n)^\top \mathbf{H}(\mathbf{x}_n) \mathbf{J}_{\hat{\mathbf{w}}}(\mathbf{x}_n), \tag{2}$$

where $\mathbf{H}(\mathbf{x}) = -\partial^2_{f_{\hat{\mathbf{w}}}(\mathbf{x})} \log p(\mathbf{y}|f_{\hat{\mathbf{w}}}(\mathbf{x})) \in \mathbb{R}^{O \times O}$ is the Hessian of the log-likelihood and we have assumed a weight prior $\mathcal{N}(\mathbf{0}, \alpha^{-1}\mathbf{I})$. Note that it is trivial to extend to other prior covariances. This particular covariance is known as the *generalized Gauss-Newton* (GGN) Hessian approximation, which is commonly used in Laplace approximations (Daxberger et al., 2021b).

To reduce the notational load we stack the per-datum Jacobians into $\mathbf{J}_{\hat{\mathbf{w}}} = [\mathbf{J}_{\hat{\mathbf{w}}}(\mathbf{x}_1); \ldots ; \mathbf{J}_{\hat{\mathbf{w}}}(\mathbf{x}_N)] \in \mathbb{R}^{NO \times D}$ and similarly for the Hessians, and write the GGN matrix as $\text{GGN}_{\hat{\mathbf{w}}} = \mathbf{J}_{\hat{\mathbf{w}}}^\top \mathbf{H} \mathbf{J}_{\hat{\mathbf{w}}}$. For Gaussian likelihoods, the Hessian is an identity matrix and can be disregarded, and for other likelihoods simple expressions are generally available (Immer et al., 2021b).

**Sampled and linearized Laplace.**   The Laplace approximation gives a Gaussian distribution $q(\mathbf{w}|\mathcal{D})$ over the weight space with mean $\hat{\mathbf{w}}$ and covariance $\Sigma$. A predictive distribution is obtained by integrating the approximate posterior against the model likelihood,

$$p(\mathbf{y}^*|\mathbf{x}^*, \mathcal{D}) = \mathbb{E}_{\mathbf{w} \sim q}[p(\mathbf{y}^*|f(\mathbf{w}, \mathbf{x}^*))] \approx \frac{1}{S} \sum_{i=1}^{S} p(\mathbf{y}^*|f(\mathbf{w}_i, \mathbf{x}^*)), \quad \mathbf{w}_i \sim q. \tag{3}$$

We refer to this predictive method as *sampled Laplace*. Recent works have suggested *linearizing* the neural network in the likelihood model to obtain the predictive distribution (Immer et al., 2021b),

$$p(\mathbf{y}^*|\mathbf{x}^*, \mathcal{D}) = \mathbb{E}_{\mathbf{w} \sim q}[p(\mathbf{y}^*|f_{\text{lin}}^{\hat{\mathbf{w}}}(\mathbf{w}, \mathbf{x}^*))] \approx \frac{1}{S} \sum_{i=1}^{S} p(\mathbf{y}^*|f_{\text{lin}}^{\hat{\mathbf{w}}}(\mathbf{w}_i, \mathbf{x}^*)), \quad \mathbf{w}_i \sim q. \tag{4}$$

This is referred to as *linearised Laplace*. Immer et al. (2021b) argues that the common choice of approximating the posterior precision with the GGN implicitly linearizes the neural network and hence the predictive distribution should be modified for consistency.

Sampled Laplace is known to severely *underfit*, whereas the linearized Laplace approximation does not (Immer et al. 2021b; Fig. 2). It is an open problem *why* the crude linearization is beneficial (Papamarkou et al., 2024). This paper shows that the benefit is linked to the lack of *reparameterization invariance*.

**The lack of reparameterization invariance** leads to an additional problem for Laplace approximations. The precision of the approximate posterior is given either by the Hessian or the GGN. As shown by Dinh et al. (2017), the Hessian of the loss is not invariant to reparameterizations of the neural network, and the same holds for the GGN. Depending on which parametrization of the posterior mode is chosen by the optimizer, we, thus, get different covariances for the approximate posterior. Empirically, this can render Laplace's approximation unstable (Warburg et al., 2023). Figure 1 (center) illustrates the phenomena.

## 3   Reparameterizations of linear functions

Deep learning models excel when they are highly *overparametrized*, i.e. when they have significantly more parameters than observations ($D \gg NO$). This introduces many degrees of freedom to the model, which will be reflected in the Bayesian posterior. However, as we have argued, traditional approximate Bayesian inference does not correctly capture this and assigns different probability measures to identical functions. Next, we characterize these degrees of freedom to design suitable approximate posteriors. To develop the theory, we first consider the linear setting and then extend it to the general case.

**The reparameterizations of linear functions** can be characterized exactly.    Consider $f(\mathbf{w}) = \mathbf{A}\mathbf{w} + \mathbf{b}$ and a possible reparameterization, $g : \mathbb{R}^D \to \mathbb{R}^D$, of this function such that $f(g(\mathbf{w})) = f(\mathbf{w})$. It is then evident that $\mathbf{A}(g(\mathbf{w}) - \mathbf{w}) = \mathbf{0}$. This implies that for any reparameterization of a linear function, we have $g(\mathbf{w}) - \mathbf{w} \in \ker(\mathbf{A})$, where $\ker(\mathbf{A})$ denotes the *kernel* (nullspace) of $\mathbf{A}$. Hence, the linear function cannot be reparametrized if we restrict ourselves to the non-kernel subspace of the input space or if $\mathbf{A}$ has a trivial kernel.

**A linearized neural network** $f_{\text{lin}}^{\mathbf{w}'} : \mathbf{w}, \mathbf{x} \mapsto f_{\mathbf{w}'}(\mathbf{x}) + \mathbf{J}_{\mathbf{w}'}(\mathbf{x})(\mathbf{w} - \mathbf{w}')$ is a linear function in the parameters, where we have linearized around $\mathbf{w}'$. The above analysis then implies that the kernel of the stacked Jacobian $\mathbf{J}_{\mathbf{w}'}$ characterizes the reparameterizations of the linearized network.

We can also characterize the reparameterizations through the GGN and the corresponding *neural tangent kernel* (NTK; Jacot et al. 2018),

$$\mathrm{GGN_w} = \mathbf{J}_{\mathbf{w}}^{\top}\mathbf{J}_{\mathbf{w}}, \qquad \mathrm{NTK_w} = \mathbf{J}_{\mathbf{w}}\mathbf{J}_{\mathbf{w}}^{\top}. \tag{5}$$

By construction, these have the same non-zero eigenvalues, and thereby also have identical ranks. We, thus, see that the kernel of the Jacobian coincides with that of the GGN, i.e. $\ker(\mathbf{J}_{\mathbf{w}}) = \ker(\mathrm{GGN_w})$.

**Two orthogonal subspaces.** For any self-adjoint operator (such as positive semi-definite matrices like the GGN), the *image* and the *kernel* orthogonally span the whole space, i.e.

$$\mathrm{im}(\mathrm{GGN_w}) \oplus \ker(\mathrm{GGN_w}) = \mathbb{R}^D, \quad (6)$$

where the *kernel* is the hyperplane of vectors that are mapped to zero and the *image* is the hyperplane of vectors spanned by the operator (Fig. 3). For a linearized neural network, $\mathrm{im}(\mathrm{GGN_w})$ spans the *effective* parameters $\mathcal{P} \subset \mathbb{R}^D$, i.e. the maximal set of parameters that generate different linear functions $\mathbb{R}^I \to \mathbb{R}^O$ when evaluated on the training set.

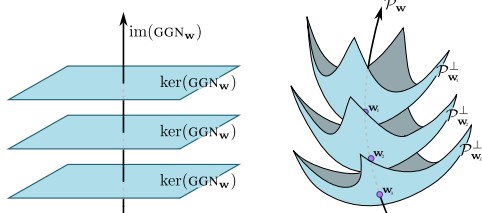

Figure 3: The weight space can be decomposed into directions of *reparameterizations* and *functional changes*. For linear models (left) these are linear subspaces given by the kernel and the image, respectively. For nonlinear models, these are the nonlinear manifolds $\mathcal{P}^{\perp}_{\mathbf{w}_i}$ and $\mathcal{P}_{\mathbf{w}_i}$, respectively.

**A Laplace covariance decomposes** into the same subspaces. Recall that the posterior precision is $\Sigma^{-1} = \mathrm{GGN_{\hat{w}}} + \alpha\mathbf{I}$. Let the eigendecomposition of $\mathrm{GGN_{\hat{w}}}$ be $\mathbf{U}^T\mathbf{\Lambda}\mathbf{U}$, and assume that $\mathbf{U_1}$ and $\mathbf{U_2}$ are the eigenvectors corresponding to the non-zero eigenvalues $\tilde{\mathbf{\Lambda}}$, and the zero eigenvalues respectively. These form a basis in the kernel and image subspace as discussed above. Then the covariance is,

$$\Sigma = \left( \left[ \frac{\mathbf{U_1}}{\mathbf{U_2}} \right]^{T} \left[ \begin{array}{c|c} \tilde{\mathbf{\Lambda}} & \mathbf{0} \\ \hline \mathbf{0} & \mathbf{0} \end{array} \right] \left[ \frac{\mathbf{U_1}}{\mathbf{U_2}} \right] + \alpha\mathbf{I} \right)^{-1} = \mathbf{U_1^T}(\tilde{\mathbf{\Lambda}} + \alpha\mathbf{I}_k)^{-1}\mathbf{U_1} + \alpha^{-1}\mathbf{U_2^T}\mathbf{U_2}. \tag{7}$$

Consequently, we can decompose any sample from the Gaussian $\mathcal{N}(\hat{\mathbf{w}}, \Sigma)$ into a kernel and an image contribution, $\mathbf{w} = \hat{\mathbf{w}} + \mathbf{w}_{\mathrm{ker}} + \mathbf{w}_{\mathrm{im}}$, where $\mathbf{w}_{\mathrm{ker}}$ is the component of the sample that is in the kernel of $\mathrm{GGN_{\hat{w}}}$ and $\mathbf{w}_{\mathrm{im}}$ is in the image. Note that all probability mass in $\ker(\mathrm{GGN_{\hat{w}}})$ is due to the prior, i.e. we place prior probability on functional reparameterizations even if we can never observe data in support of such.

**Underfitting in sampled Laplace** can now be understood. For the linearized approximation, it holds for training data $\mathbf{x} \in \mathcal{X}$ that,

$$f_{\mathrm{lin}}^{\hat{\mathbf{w}}}(\hat{\mathbf{w}} + \mathbf{w}_{\mathrm{ker}} + \mathbf{w}_{\mathrm{im}}, \mathbf{x}) = f_{\mathrm{lin}}^{\hat{\mathbf{w}}}(\hat{\mathbf{w}} + \mathbf{w}_{\mathrm{im}}, \mathbf{x}).$$

Hence, the linearized predictive only samples in the image subspace consisting of unique functions. This is *not* true for the sampled Laplace approximation, which also samples in the kernel subspace. Since sampled Laplace does not linearize the neural network, the kernel does not correspond to reparameterizations. It hence adds "incorrect" degrees of freedom to the posterior as artifacts of the Gaussian approximation.

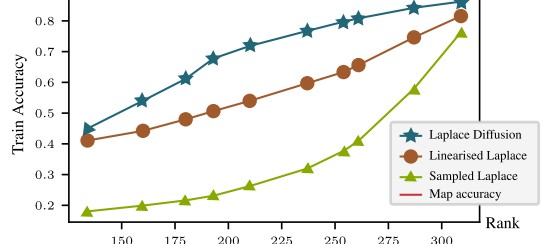

Figure 4: Underfitting of sampled Laplace is less pronounced when the rank of the GGN is higher for a fixed number of parameters. This is consistent with our hypothesis as a high GGN rank implies a lower dimensional kernel. For experimental details, see appendix E.2.

Empirically, sampled Laplace is only observed to underfit in overparametrized models. Fig. 4 illustrates this by increasing the amount of training data to decrease the kernel rank, i.e. reduce the reparametrization issue. We find that as the issue is lessened, sampled Laplace reduces its underfitting.

## 4 Reparameterizations of neural networks

We have seen that the parameters of linear models can be decomposed into two linear subspaces corresponding to reparameterizations and functional changes. We next analyze nonlinear models.

**Intuitively**, reparameterizations of a nonlinear neural network form continuous trajectories in the parameter space (c.f. Fig. 1). We define that all points along such a trajectory are identical, which changes the weight space geometry to be a manifold. Likewise, the parameter changes corresponding to actual function changes reside on a nonlinear manifold. This is sketched in Fig. 3. Interestingly, the GGN turns out to induce a natural (local) inner product on these nonlinear manifolds, which allows us to both understand and generalize the linearized Laplace approximation.

## 4.1 The effective-parameters quotient space

For a *nonlinear* neural network $f : \mathbb{R}^D \times \mathbb{R}^I \to \mathbb{R}^O$, the surfaces in weight space along which the function does not change are generally *not* linear. Here, we formalize these reparameterization invariant surfaces and show that they are a partition of the weight space.

**Definition 4.1.** Given a datapoint $\mathbf{x} \in \mathbb{R}^I$, for any $\mathbf{w} \in \mathbb{R}^D$ we define the $\mathbf{x}$-*reparameterizations* as the set $\mathcal{R}_{\mathbf{x}}^f(\mathbf{w}) = \{\mathbf{w}' \text{ such that } f(\mathbf{w}', \mathbf{x}) = f(\mathbf{w}, \mathbf{x})\}$. Consistently, given a collection of points $\mathcal{X} \subseteq \mathbb{R}^I$, we call the intersection $\mathcal{R}_{\mathcal{X}}^f(\mathbf{w}) = \bigcap_{\mathbf{x} \in \mathcal{X}} \mathcal{R}_{\mathbf{x}}^f(\mathbf{w})$ $\mathcal{X}$-*reparameterizations*.

Trivially, $\mathbf{w} \in \mathcal{R}_{\mathcal{X}}^f(\mathbf{w})$ for any choice of $\mathcal{X}$. We next define the subset of $\mathcal{X}$-reparameterizations which can be obtained via a smooth deformation from $\mathbf{w}$.

**Definition 4.2.** We say that a piecewise differentiable function $\gamma : [0, 1] \to \mathbb{R}^D$ is a *homotopy* of $(\mathbf{w}, \hat{\mathbf{w}})$ if $\gamma(0) = \mathbf{w}$ and $\gamma(1) = \mathbf{w}'$. The set of $\mathcal{X}$-*smooth-reparameterizations* is defined as,

$$\bar{\mathcal{R}}_{\mathcal{X}}^f(\mathbf{w}) = \left\{ \mathbf{w}' \text{ such that } \begin{array}{l} \exists \gamma \text{ a homotopy of } (\mathbf{w}, \mathbf{w}') \\ \gamma(t) \in \mathcal{R}_{\mathcal{X}}^f(\mathbf{w}) \; \forall t \in [0, 1] \end{array} \right\}.$$

A homotopy $\gamma$ is, thus, a smooth path along which all neural networks have identical predictions on $\mathcal{X}$. We consider two networks, $\mathbf{w}$ and $\mathbf{w}'$, similar if they can be connected by such a homotopy. Formally, we define the relation $\sim$ over $\mathbb{R}^D$ as $\mathbf{w} \sim \mathbf{w}'$ if $\mathbf{w}' \in \bar{\mathcal{R}}_{\mathcal{X}}^f(\mathbf{w})$.

We next use this relation to form a new view on the weight space $\mathbb{R}^D$ in which similar weights are seen as *one* point. This can be realized using *quotient spaces* (Lee, 2012). These are well-studied spaces that are constructed by considering a collection of points in one space as a single point in a new space. In our case, we have the following result.

**Lemma 4.3.** $\sim$ *is an equivalence relation, i.e. it is transitive, symmetric and reflexive. We can form the quotient space* $\mathcal{P} = \mathbb{R}^D / \sim$ *of effective parameters. We denote* $[\mathbf{w}] \in \mathcal{P}$ *the equivalence class of an element* $\mathbf{w} \in \mathbb{R}^D$.

This quotient structure gives a rich mathematical foundation to construct reparameterization invariant neural networks. Within the quotient, two effective parameters $[\mathbf{w}_1], [\mathbf{w}_2] \in \mathcal{P}$ are the same point if and only if $\mathbf{w}_1 \sim \mathbf{w}_2$. This means that all parameters $\mathbf{w} \in [\mathbf{w}_1]$ gives the same function over $\mathcal{X}$.

## 4.2 The effective-parameters manifold

Geometry is the mathematical language of invariances. To this end would like to endow the weight space with a geometric structure such that two weights, $\mathbf{w}_1$ and $\mathbf{w}_2$, corresponding to the same function, have a distance of zero, i.e.

$$\text{dist}(\mathbf{w}_1, \mathbf{w}_2) = 0 \quad \Leftrightarrow \quad \mathbf{w}_1 \sim \mathbf{w}_2. \tag{8}$$

Since the weights generate the same function, we define a metric that measures differences in *function values* on the training data. Consider weights $\mathbf{w}$ and an infinitesimal displacement $\boldsymbol{\epsilon}$, we then define,

$$\text{dist}^2(\mathbf{w}, \mathbf{w} + \boldsymbol{\epsilon}) = \sum_{n=1}^N \|f(\mathbf{w}, \mathbf{x}_n) - f(\mathbf{w} + \epsilon, \mathbf{x}_n)\|^2 = \boldsymbol{\epsilon}^\top \text{GGN}_{\mathbf{w}} \boldsymbol{\epsilon} + \mathcal{O}(\epsilon^3), \tag{9}$$

where the last step follows from a first-order Taylor expansion of $f$ around $\mathbf{w}$. This is a standard *pullback metric* $(f^*H)_{\mathbf{w}} = \text{GGN}_{\mathbf{w}}$ commonly used in Riemannian geometry. This implies that the GGN matrix infinitesimally defines an inner product, i.e. it is a *Riemannian metric*. By integrating over paths, the distance extend to any pair of points and satisfies Eq. 8 (Lee, 2012).

**Watch out! It's a pseudo-metric.** We have already seen that in overparametrized models, the GGN is rank-deficient, which implies that it is not positive definite. Consequently, it is not a Riemannian metric but rather a *pseudo*-Riemannian metric. A pseudo-metric can be a counterintuitive object: two points $\mathbf{w}_1$ and $\mathbf{w}_2$ at distance zero may have different pseudo-metrics $(f^*H)_{\mathbf{w}_1} \neq (f^*H)_{\mathbf{w}_2}$. This is reflected in the Laplace approximation. The covariance prescribed by the Laplace approximation is $\Sigma_{\hat{\mathbf{w}}} = (\nabla^2_{\mathbf{w}}\mathcal{L}(\hat{\mathbf{w}}) + \alpha\mathbf{I})^{-1}$, where $\mathcal{L}(\mathbf{w})$ is shorthand for the training log-likelihood. The Hessian is exactly the pullback pseudo-metric $\nabla^2_{\mathbf{w}}\mathcal{L}(\hat{\mathbf{w}}) = (f^*H)_{\hat{\mathbf{w}}}$, which is *not* invariant to reparameterizations of the neural network. Specifically, for a reparameterization function $g$ that is also a diffeomorphism, the change of variable rules states that,

$$\underbrace{\nabla^2_{\mathbf{w}}\mathcal{L}(g(\hat{\mathbf{w}}))}_{\Sigma^{-1}_{g(\hat{\mathbf{w}})} - \alpha\mathbf{I}} = \nabla_{\mathbf{w}}g(\hat{\mathbf{w}})^\top \underbrace{\nabla^2_{\mathbf{w}}\mathcal{L}(\hat{\mathbf{w}})}_{\Sigma^{-1}_{\hat{\mathbf{w}}} - \alpha\mathbf{I}} \nabla_{\mathbf{w}}g(\hat{\mathbf{w}}). \tag{10}$$

This means that, while each parameter $\mathbf{w}$ has its well-defined covariance $\Sigma_{\mathbf{w}}$, each equivalence class does not have a unique one, since $\hat{\mathbf{w}}$ and $g(\hat{\mathbf{w}})$ belong to the same equivalence class and $\Sigma_{\hat{\mathbf{w}}} \neq \Sigma_{g(\hat{\mathbf{w}})}$.

**Non-Gaussian likelihoods.** The Euclidean distance measure in Eq. 9 corresponds to choosing a Gaussian likelihood. The distance definition readily extends to other likelihoods and the corresponding metric takes the form of the generalized Gauss-Newton matrix $\mathbf{J}^\top_{\mathbf{w}}\mathbf{H}\mathbf{J}_{\mathbf{w}}$, where $\mathbf{H}$ denotes the Hessian of the log-likelihood. For both Gaussian and Bernoulli likelihoods, this Hessian is positive definite, but e.g. the cross entropy has a rank-deficient Hessian and, thus, induces a pseudo-metric.

**An impractical solution.** The unfortunate behavior of approximate posteriors assigning different probabilities to the same function could be rectified by marginalizing over the set of reparameterizations of $\mathbf{w}$, i.e. $\int_{\mathbf{w}' \in \mathcal{R}(\mathbf{w})} q(\mathbf{w}'|\mathcal{D})\mathrm{d}\mathbf{w}'$. While this construction solves the highlighted problem, its complexity makes it impractical and we are unaware of any works along these lines.

When restricted to a smaller class of reparameterization (the ones homotopic to the identity), the integral can be thought of as "collapsing" each reparameterization equivalence class to a single point in $\mathcal{P} = \mathbb{R}^D/\sim$ formalized in Lemma 4.3. Nontrivially, the pullback metric implicitly performs a similar operation, as shown later in Theorem 4.5. This connection motivates the dive into Riemannian geometry: *we get a tractable approach to engaging with neural network reparameterizations*.

### 4.3 Topological equivalence of the two views

So far we described two *a priori* very different objects: the quotient space $\mathcal{P} = \mathbb{R}^D/\sim$ and the pseudo-Riemannian manifold $(\mathbb{R}^D, \mathrm{GGN}_{\mathbf{w}})$. We referred to both of them as *effective parameters* and this is no coincidence as there is a natural relationship between the points at distance zero according to the pseudo-metric and the equivalence classes.

**Proposition 4.4.** *For any $\mathbf{w}_0, \mathbf{w}_1 \in \mathbb{R}^D$ it holds*

$$d_{f^*H}(\mathbf{w}_0, \mathbf{w}_1) = 0 \quad \Longleftrightarrow \quad [\mathbf{w}_0] = [\mathbf{w}_1] \in \mathcal{P}. \tag{11}$$

Even better, these two spaces share the same topological structure. To state this we need a notion of distance on the quotient space and the most natural choice is to inherit the Euclidean distance $\|\cdot\|$ from $\mathbb{R}^D$. This distance is defined as

$$d_{\mathcal{P}}([\mathbf{w}], [\mathbf{w}']) = \inf\{\|p_1 - q_1\| + \ldots + \|p_n - q_n\|\},$$

where the infimum is taken over all finite sequences $p_1, \ldots, p_n$ and $q_1, \ldots, q_n$ such that $[\mathbf{w}] = [p_1]$, $[p_{i+1}] = [q_i]$ and $[q_n] = [\mathbf{w}']$.

This distance $d_{\mathcal{P}}$ induces a topology on the quotient space $\mathcal{P}$ which is equivalent to the topology induced by the pullback distance $d_{f^*H}$ on the pseudo-Riemannian manifold. Formally

**Theorem 4.5.** *For any $\mathbf{w}_0, \mathbf{w}_1 \in \mathbb{R}^D$, for any $\epsilon > 0$ there exists $\delta > 0$ such that*

$$d_{\mathcal{P}}([\mathbf{w}_0], [\mathbf{w}_1]) < \delta \quad \Longrightarrow \quad d_{f^*H}(\mathbf{w}_0, \mathbf{w}_1) < \epsilon \tag{12}$$

$$d_{f^*H}(\mathbf{w}_0, \mathbf{w}_1) < \delta \quad \Longrightarrow \quad d_{\mathcal{P}}([\mathbf{w}_0], [\mathbf{w}_1]) < \epsilon. \tag{13}$$

This result connects an abstract quotient space $\mathcal{P}$ with the pseudo-Riemannian metric $\mathrm{GGN}_{\mathbf{w}}$. The quotient captures useful intuitions but is difficult to leverage computationally. In contrast, the pseudo-metric has some counterintuitive aspects but we can identify the underlying Riemannian structure which leads to tractable algorithms (Sec. 5).

**A tale of two manifolds.** For any given parameter $\mathbf{w} \in \mathbb{R}^D$ and training set $\mathcal{X}$, we show that there exist two Riemannian manifolds $(\mathcal{P}_\mathbf{w}, \mathfrak{m})$ and $(\mathcal{P}_\mathbf{w}^\perp, \mathfrak{m}^\perp)$ embedded in $\mathbb{R}^D$, illustrated in Fig. 3. They capture the functional change and reparameterization properties respectively, but, differently from the previously studied $(\mathbb{R}^D, \mathrm{GGN}_\mathbf{w})$, they are Riemannian manifolds without degenerate directions in their metrics. Formally,

**Theorem 4.6.** *For any parameter* $\mathbf{w}$ *suppose the set of parameters that generate the same predictions is denoted by* $\mathcal{P}_\mathbf{w}^\perp = \{\mathbf{w}' \in \mathbb{R}^D$ *such that* $f(\mathbf{w}', x) = f(\mathbf{w}, x)$ *for all* $x \in \mathcal{X}\}$*. Then this set is a smooth manifold embedded in* $\mathbb{R}^D$*. Furthermore, the set of parameters that locally generates unique predictions,* $\mathcal{P}_\mathbf{w}$ *is also a submanifold embedded in* $\mathbb{R}^D$*.*

They are the direct generalization to the nonlinear case of the two spaces involved in Eq. 6, where $\mathcal{P}_\mathbf{w}$ plays the role of the *image* and $\mathcal{P}_\mathbf{w}^\perp$ plays the role of the *kernel*. When $f$ is linear, they are identical.

In general, $\mathcal{P}_\mathbf{w}$ and $\mathcal{P}_\mathbf{w}^\perp$ intersect only in $\mathbf{w}$, and the two respective tangent spaces in $\mathbf{w}$ span all directions. They can be thought of as two collections of parameters, and the associated functions have different properties: (1) $\mathcal{P}_\mathbf{w}^\perp$ is entirely contained in the same equivalence class $\mathcal{P}_\mathbf{w}^\perp \subseteq [\mathbf{w}]$, thus all the parametrized functions are identical on the train set; in contrast, (2) $\mathcal{P}_\mathbf{w}$ never intersects the same equivalence class more than one time, at least locally, thus the parametrized functions always changes when moving in any direction. Thus $\mathcal{P}_\mathbf{w}$ resembles the effective parameter manifold $\mathcal{P}$, but with the difference of being an actual Riemannian manifold. These two manifolds exist under the assumption that Jacobian is full rank (see proof in appendix C).

The two metrics $\mathfrak{m}$ and $\mathfrak{m}^\perp$ are not uniquely defined. A natural choice for $\mathfrak{m}$ is to restrict $\mathrm{GGN}_\mathbf{w}$ to the tangent space of $\mathcal{P}_\mathbf{w}$ corresponding to the non-zero eigenvectors, i.e. $\mathfrak{m} = \mathrm{GGN}_\mathbf{w}^+$. While, for $\mathfrak{m}^\perp$ we can inherit the Euclidean metric, i.e. $\mathfrak{m}^\perp = \alpha \mathbf{I}$ for $\alpha > 0$.

# 5   Exploring manifolds with random walks

**SDEs on manifolds.** Given a Riemannian manifold $(\mathrm{M}, \mathbf{G})$, the simplest choice of distribution that respects the Riemannian metric $\mathbf{G}$ is a Riemannian diffusion (or Brownian motion, c.f. Hsu (2002)) stopped at time $t$. This follows the stochastic differential equation (Girolami & Calderhead, 2011),

$$d\mathbf{w} = \sqrt{2\tau}\mathbf{G}(\mathbf{w})^{-\frac{1}{2}}dW + \tau\Gamma dt \qquad \text{where} \quad \Gamma_i(\mathbf{w}) = \sum_{j=1}^{D} \frac{\partial}{\partial \mathbf{w}_j}(\mathbf{G}(\mathbf{w})^{-1})_{ij}. \tag{14}$$

Practically speaking this simple process can be simulated using an Euler–Maruyama (Maruyama, 1955) scheme. The Christoffel symbols, $\Gamma_i(\boldsymbol{\theta})$, are commonly disregarded as they have a high computational cost, and Li et al. (2015) showed that the resulting error is bounded.

Using the Euler–Maruyama integrator with step size $h_t$, setting $\tau = 1$ corresponding to standard Bayesian inference and disregarding the term involving the Christoffel symbols $\Gamma$, we obtain the simple update rule $\mathbf{w}_{t+1} = \mathbf{w}_t + \sqrt{2h_t}\mathbf{G}(\mathbf{w}_t)^{-\frac{1}{2}}\boldsymbol{\epsilon}$, where $\boldsymbol{\epsilon} \sim \mathcal{N}(\mathbf{0}, \mathbf{I})$. This applies to any Riemannian manifold. However, the effective-parameter $(\mathbb{R}^D, \mathrm{GGN}_\mathbf{w})$ is only pseudo-Riemannian, we explore three Riemannian alternatives: $(\mathbb{R}^D, \mathrm{GGN}_\mathbf{w} + \alpha\mathbb{I})$, $(\mathcal{P}_\mathbf{w}^\perp, \alpha\mathbf{I})$ and $(\mathcal{P}_\mathbf{w}, \mathrm{GGN}_\boldsymbol{\omega}^+)$

**Diffusion on** $(\mathbb{R}^D, \mathrm{GGN}_\mathbf{w} + \alpha\mathbf{I})$. The Laplace approximation can also be written as a diffusion on a manifold. As we saw in Sec. 2, the Laplace approximation can be written as $\mathbf{w}|\mathcal{D} \sim \mathcal{N}(\hat{\mathbf{w}}, \Sigma)$ with $\Sigma^{-1} = \mathrm{GGN}_{\hat{\mathbf{w}}} + \alpha\mathbf{I}$. This can also be written as a sample at $t = 1$ of a Riemannian diffusion on a manifold with a constant metric, $(\mathbb{R}^D, \mathbf{G})$, where $\mathbf{G} = \mathrm{GGN}_{\hat{\mathbf{w}}} + \alpha\mathbf{I}$. The SDE $d\mathbf{w} = \mathbf{G}^{-\frac{1}{2}}dW$ have a marginal distribution at $t = 1$ that exactly match the standard Laplace approximation. Note that this formulation does not rely on the approximation of the SDE that disregards the term involving the Christoffel symbols $\Gamma$ as these are zero for constant metrics. Hence, the above is exactly a Riemannian diffusion on the manifold with a constant metric given by the GGN at the MAP parameter. Note that this is only a valid diffusion for $\alpha > 0$ in which case it is not reparametrization invariant.

**Kernel-manifold diffusion.** The kernel-manifold $(\mathcal{P}_\mathbf{w}^\perp, \alpha\mathbf{I})$ consists of parameters that generate the same function over the training set. The effect of diffusion on this manifold and using the neural network predictive is similar to sampling from the kernel subspace while using the linearized predictive. On the training set the predictive variance is $0$ because it only samples reparametrizations of the MAP predictions. On out-of-distribution data, the variance is greater than $0$ if at least one of

Table 1: In-distribution performance across methods trained on MNIST, FMNIST and CIFAR-10.

| | | Conf. ($\uparrow$) | NLL ($\downarrow$) | Acc. ($\uparrow$) | Brier ($\downarrow$) | ECE ($\downarrow$) | MCE ($\downarrow$) |
|---|---|---|---|---|---|---|---|
| MNIST | Laplace Diffusion (ours) | **0.988±0.001** | **0.042±0.007** | **0.987±0.002** | **0.022±0.003** | **0.137±0.019** | **0.775±0.043** |
| | Sampled Laplace | 0.589±0.008 | 3.812±0.284 | 0.146±0.032 | 1.176±0.046 | 0.443±0.026 | 0.985±0.002 |
| | Linearised Laplace | 0.968±0.004 | 0.306±0.041 | 0.926±0.008 | 0.117±0.012 | 0.251±0.034 | 0.855±0.041 |
| FMNIST | Laplace Diffusion (ours) | **0.900±0.001** | **0.001±0.000** | **0.906±0.007** | **0.141±0.006** | **0.108±0.015** | **0.729±0.092** |
| | Sampled Laplace | 0.618±0.021 | 4.507±0.000 | 0.098±0.010 | 1.295±0.014 | 0.518±0.013 | 0.986±0.001 |
| | Linearised Laplace | 0.897±0.003 | 0.423±0.000 | 0.862±0.005 | 0.207±0.006 | 0.147±0.017 | 0.756±0.048 |
| CIFAR-10 | Laplace Diffusion (ours) | **0.952±0.007** | **0.345±0.062** | **0.905±0.007** | **0.155±0.019** | 0.259±0.008 | 0.870±0.021 |
| | Sampled Laplace | 0.843±0.004 | 0.997±0.222 | 0.717±0.049 | 0.422±0.081 | **0.221±0.047** | 0.804±0.080 |
| | Linearised Laplace | 0.951±0.007 | 0.614±0.020 | 0.863±0.001 | 0.222±0.002 | 0.337±0.022 | **0.789±0.035** |

Table 2: Out-of-distribution AUROC ($\uparrow$) performance for MNIST, FMNIST and CIFAR-10.

| Trained on | MNIST | | | FMNIST | | | CIFAR-10 | |
|---|---|---|---|---|---|---|---|---|
| Tested on | FMNIST | EMNIST | KMNIST | MNIST | EMNIST | KMNIST | CIFAR-100 | SVHN |
| Laplace Diffusion (ours) | **0.909±0.033** | **0.625±0.018** | **0.929±0.008** | **0.759±0.045** | **0.741±0.010** | **0.749±0.023** | **0.851±0.002** | **0.862±0.010** |
| Sampled Laplace | 0.500±0.026 | 0.494±0.006 | 0.482±0.013 | 0.495±0.037 | 0.503±0.036 | 0.493±0.033 | 0.687±0.033 | 0.599±0.038 |
| Linearised Laplace | 0.758±0.070 | 0.602±0.027 | 0.790±0.018 | 0.625±0.050 | 0.628±0.013 | 0.624±0.020 | 0.837±0.006 | 0.854±0.024 |

the reparameterizations on the training set is not a global reparameterization. This leads to a clear separation in the predictive variance of in-distribution and out-of-distribution data (Fig. 2) and further implies that this diffusion distribution never underfits. Stated formally,

**Theorem 5.1.** $\mathrm{Var}_{\mathbf{w}\sim\mathcal{P}_{\tilde{\mathbf{w}}}^{\perp}}[f(\mathbf{w},\mathbf{x})] = 0$ *for train data* $\mathbf{x}\in\mathcal{X}$. *For a test point* $\mathbf{x}_t\notin\mathcal{X}$, *if there exists a reparameterization* $\mathbf{w}'\in\bar{\mathcal{R}}_{\mathcal{X}}^{f}(\hat{\mathbf{w}})$ *such that* $\mathbf{w}'\notin\bar{\mathcal{R}}_{\mathcal{X}\cup\{\mathbf{x}_t\}}^{f}(\hat{\mathbf{w}})$, *then* $\mathrm{Var}_{\mathbf{w}\sim\mathcal{P}_{\tilde{\mathbf{w}}}^{\perp}}[f(\mathbf{w},\mathbf{x}_t)] > 0$.

**Non-kernel-parameter manifold diffusion.** The non-kernel-parameter manifold $(\mathcal{P}_{\mathbf{w}}, \mathrm{GGN}_{\boldsymbol{\omega}}^{+})$ consists of parameters that generate unique functions over the training set. Diffusion on this manifold samples functions that are necessarily different from the MAP predictions on the training set. However, the predictive variance in the training set is bounded such that the functional diversity in the predictive samples reflects the intrinsic variance of the training data (Fig. 2).

This is the only considered diffusion that acts on a Riemannian manifold while being reparametrization invariant, i.e. $\bar{\mathcal{R}}_{\mathcal{X}}^{f}(\mathbf{w}) = \{\mathbf{w}\}$. We call this *Laplace diffusion* and study it empirically in Sec. 7.

# 6 Related work

Bayesian deep learning techniques are still in their infancy and generally involve poorly understood approximations. The arguably most popular tool for uncertainty quantification is *ensembles* (Lakshminarayanan et al., 2017; Hansen & Salamon, 1990). Several approaches make Gaussian approximations to the true posterior, including *'Bayes by backprop'* (Blundell et al., 2015), *stochastic weight averaging* (SWAG) (Maddox et al., 2019) and the *Laplace approximation* (MacKay, 1992; Daxberger et al., 2022; Antorán et al., 2023; Deng et al., 2022; Miani et al., 2022).

The high dimensionality of the weight space gives rise to significant computational challenges when constructing Bayesian approximations. This has motivated various low-rank approximations (review in Daxberger et al., 2021a), e.g. *last layer* approximations (Kristiadi et al., 2020), *subnetwork inference* (Daxberger et al., 2021c), *subspace inference* (Izmailov et al., 2020) or even PCA in weight space (Maddox et al., 2019). Such approaches lessen the computational load, while often improving predictive performance. Our analysis sheds light on why crude approximations perform favorably: *smaller models are less affected by reparameterization issues*. Our diffusion process, thus, provides an alternative, and less heuristic, path forward.

MacKay (1998) noted the importance of the choice of basis in Laplace approximations; our pseudo-Riemannian view can be seen as having a continuously changing basis. Kristiadi et al. (2023) studied how a metric transforms under a bijective differentiable change of variables. They enforce geometric

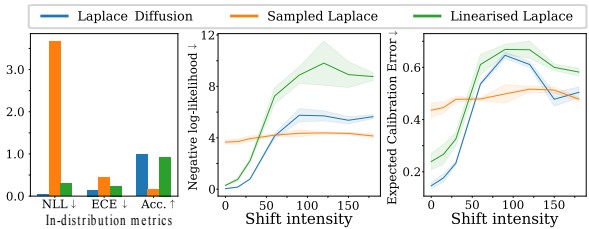

Figure 5: Benchmark results for Ro-tated MNIST (similar results for FM-NIST and CIFAR are in appendix E.3.2). Sampled Laplace significantly under-fits even for non-rotated data. Laplace diffusion consistently outperforms the other methods.

consistency, highlighting, e.g., the non-invariance of the GGN to a change of variables. Petzka et al. (2019); Jang et al. (2022) point to the same inconsistency with an emphasis on flatness measures.

Kim et al. (2022) and Antorán et al. (2022) study global (rather than data-dependant) reparametriza-tions associated with specialized architectures. While analytic expressions can be obtained, the results do not apply to general networks. While not expressed in terms of reparametrizations, Izmailov et al. (2021) show that linearly dependent datasets give rise to a hyperplane in the kernel manifold. Kim et al. (2024) also study the kernel of the GGN in the context of influence functions. These works characterize subsets of the reparametrization group. We provide the first architecture-agnostic characterization of all continuous reparametrizations.

In a closely related work, Bergamin et al. (2024) introduced a Riemannian Laplace approximation (Hauberg, 2018) that improves posterior fit over a range of tasks. Furthering this line of research, Yu et al. (2023) explored the use of the Fisher information metric within this framework. While sharing the language of Riemannian geometry, our work focuses on analyzing the effectiveness of linearized Laplace within the context of neural network reparametrization, instead of primarily aiming to achieve better posterior approximations. This allows us to gain deeper insights into the underlying mechanisms that contribute to the success of this approximation technique.

# 7 Experiments

We benchmark Laplace diffusion with neural network predictive against linearized and sampled Laplace to validate the developed theory. Implementation details are in Appendix E.1. We will show that the diffusion posterior slightly outperforms linearized Laplace in terms of both in-distribution fit and out-of-distribution detection. For completeness, we include comparisons to other baselines such as SWAG, diagonal Laplace, and last-layer Laplace in Appendix E.3. Laplace diffusion is competitive with the best-performing Bayesian methods despite using the neural network predictive (i.e. no linearization). This contrasts sampled Laplace which severely underfits. This is evidence that the developed theory explains the key challenges of Bayesian deep learning.

**Experimental details (appendix E.3).** We train a 44,000-parameter LeNet(LeCun et al., 1989) on MNIST and FMNIST as well as a 270,000-parameter ResNet(He et al., 2016) on CIFAR-10(Krizhevsky et al., 2009). We sample from the Laplace approximation of the posterior and our Laplace diffusion. For the samples from the Laplace approximation, we consider both the linearized predictive and the neural network predictive, while for diffusion samples, we only consider the neural network predictive. These baselines were chosen to be as similar as possible to our approach to ease the comparison. We use the same prior precision for all methods to ensure a fair comparison.

**In-distribution performance (Table 1).** We measure the in-distribution performance of different posteriors on a held-out test set. We report means ± standard deviations of several metrics: Confi-dence, Accuracy, Negative Log-Likelihood, Brier Score (Brier, 1950), Expected Calibration Error (Naeini et al., 2015) and Mean Calibration Error. We observe that Laplace diffusion has the best calibration and fit. We also confirm the underfitting of Sampled Laplace across cases. For CIFAR-10 we had to use a large prior precision to get meaningful samples from sampled Laplace, which explains the less severe underfitting. High prior precision is known to help with underfitting in sampled Laplace, but it also shrinks predictive uncertainty to almost zero.

**Robustness to dataset shift (Fig. 5, appendix E.3.2).** We use ROTATED-MNIST, ROTATED-FMNIST, and ROTATED-CIFAR to asses model-calibration and model fit under distribution shift. Fig. 5 plots negative log-likelihood (NLL) and expected calibration error (ECE) against the degrees of rotation. Laplace diffusion improves on other Laplace approximations.

Code: `https://github.com/h-roy/geometric-laplace`.

**Out-of-distribution detection (Table 2).** On out-of-distribution data from other benchmarks, we see that Laplace diffusion outperforms the other Laplace approximations.

# 8 Conclusion

While approximate Bayesian inference excels in many areas, it continues to face challenges in deep learning. Techniques that work well in shallow models struggle with deep ones even if they remain computationally tractable. This suggests that overparametrization plays a negative role in Bayesian models. Our theoretical analysis shows how overparametrization creates a growing reparameterization issue that conflicts with standard Euclidean approximate posteriors, such as the ever-present Gaussian. For small models this issue is negligible, but as models grow, so does the reparameterization issue.

Our geometric analysis also suggests a solution: we should consider approximate posteriors that respect the group structure of the reparameterizations. We observe that the generalized Gauss-Newton (GGN) matrix commonly used in Laplace approximations induces a pseudo-Riemannian structure on the parameter space that respects the topology of the reparameterization group. This implies that we can use pseudo-Riemannian probability distributions as approximate posteriors, and we experimented with the obvious choice of a geometric diffusion process. We also showed that the state-of-the-art *linearized* Laplace approximation can be viewed as a naïve (or simple) numerical approximation to our proposed diffusion. This helps explain the success of the linearized approximation.

Our proposed approximate posterior does have issues. While sampling has the same complexity as standard Laplace approximations, it increases runtime by a constant factor. Common Laplace approximations do not sample according to the GGN but rather approximate this matrix with a diagonal or block-diagonal matrix. Mathematically, such approximations break the motivational reparameterization invariance, so it is unclear if such approaches should be applied in our framework. Our work, thus, raises the need for new computational pipelines for engaging with the GGN matrix.

## Acknowledgments and Disclosure of Funding

This work was supported by a research grant (42062) from VILLUM FONDEN. This project received funding from the European Research Council (ERC) under the European Union's Horizon 2020 research and innovation programme (grant agreements 757360 and 101123955). The work was partly funded by the Novo Nordisk Foundation through the Center for Basic Machine Learning Research in Life Science (NNF20OC0062606). In addition to the ERC (above), PH, MP and LT thank the International Max Planck Research School for Intelligent Systems (IMPRS-IS) for support, and gratefully acknowledge financial support by the DFG Cluster of Excellence "Machine Learning - New Perspectives for Science", EXC 2064/1, project number 390727645; the German Federal Ministry of Education and Research (BMBF) through the Tübingen AI Center (FKZ: 01IS18039A); and funds from the Ministry of Science, Research and Arts of the State of Baden-Württemberg. The authors are also grateful to Magnus Waldemar Hoff Harder for alerting us to imprecisions in an early draft of this manuscript.

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

# A Recap

**Notation.** Consider a function $f : \mathbb{R}^D \times \mathbb{R}^I \to \mathbb{R}^O$ with Jacobian $\mathbf{J_w}(\mathbf{x}) = \partial_{\mathbf{w}} f_{\mathbf{w}}(\mathbf{x})|_{\mathbf{w}=\mathbf{w}} \in \mathbb{R}^{O \times D}$ with respect to $\mathbf{w}$ evaluated in $\mathbf{x}$ and $\mathbf{w}$. For a given log-likelihood we define the Hessian w.r.t. to the output $\mathbf{H_w}(\mathbf{x}) = -\partial^2_{f_{\mathbf{w}}(\mathbf{x})} \log p(\mathbf{y}|f_w(\mathbf{x})) \in \mathbb{R}^{O \times O}$ and we assume it not to be dependent on $\mathbf{y}$ (which is true, for example, for exponential families).

Consider being given a dataset of finite size $N$, here we do not care about labels and we only refer to the collections of the datapoints $\mathcal{X} = \{\mathbf{x}_1, \ldots, \mathbf{x}_N\} \subset \mathbb{R}^I$.

Consider the stacking of the per-datum Jacobians $\mathbf{J_w} = [\mathbf{J_w}(\mathbf{x}_1); \ldots; \mathbf{J_w}(\mathbf{x}_N)] \in \mathbb{R}^{NO \times D}$, where we dropped the dependence on $\mathcal{X}$. Similarly consider $\mathbf{H_w} = \mathrm{diag}(\mathbf{H_w}(\mathbf{x}_1); \ldots; \mathbf{H_w}(\mathbf{x}_N)) \in \mathbb{R}^{NO \times NO}$ the block diagonal stacking of the Hessians.

Consider the Generalized Gauss-Newton (GGN) and the Neural Tangent Kernel (NTK) matrices

$$\mathrm{GGN_w} = \mathbf{J_w^\top H_w J_w} \in \mathbb{R}^{D \times D} \qquad \mathrm{NTK_w} = \mathbf{H_w^{1/2} J_w J_w^\top H_w^{1/2}} \in \mathbb{R}^{NO \times NO}. \tag{15}$$

Recall also that the pullback pseudo-metric is $(f^*H)_{\mathbf{w}} = \mathrm{GGN_w}$.

**Assumptions.** We assume *uniform* upper and lower bound on the eigenvalues of the NTK matrix, that is

$$\exists l, L \in \mathbb{R} \quad \text{such that} \quad 0 < l \leq \frac{\|\mathrm{NTK_w}v\|}{\|v\|} \leq L \quad \forall v \in \mathbb{R}^{NO}, \forall \mathbf{w} \in \mathbb{R}^D, \tag{16}$$

where uniform means uniform over parameters, i.e. the bounds $l, L$ holds for every $\mathbf{w}$. Moreover we assume that the Jacobian function $\mathbf{w} \mapsto J_{\mathbf{w}}(\mathbf{x})$ is Lipschitz for every $\mathbf{x} \in \mathcal{X}$.

**How unreasonable are the assumptions?** The assumption of an upper bound $L$ is equivalent to assuming that $\mathbf{w} \mapsto f(\mathbf{w}, \mathbf{x})$ is Lipschitz for each datapoint $\mathbf{x} \in \mathcal{X}$. This is true with all standard activation functions if we restrict the parameter space to a ball of fixed radius.

The assumption of a lower bound $l$ is strongly supported by the literature on NTK, thanks to its direct implications on memorization capacity and generalization. The general trend is that the more overparametrized the network is, the stronger such lower bounds are. In the hardest setting of minimum overparametrization, Bombari et al. (2022) proved a bound that holds with high probability for fully connected MLPs at initialization. Similar results hold with bigger overparametrizations Nguyen et al. (2021); Allen-Zhu et al. (2019); Du et al. (2019). Building on top of that, other lines of work Liu et al. (2020); Oymak & Soltanolkotabi (2019, 2020) proved that the NTK does not change too much during training thanks to the PL inequality framework, and in particular the proximity of the neural network dynamics to the one described by NTK, is supported by spectral bounds on the Hessian of the landscape.

Lastly, the Lipschitzness assumption on the Jacobian is potentially the most unrealistic, although it would hold, for example, if the derivatives of activations are Lipschitz and the parameters are restricted to a finite radius ball. Nonetheless, we emphasize that this assumption is only used to control that the kernel of the GGN does not "rotate too fast", which is a much weaker assumption, but also much more cluttered to state formally.

# B Proof for equivalence of the two settings

This section contains the proof of Proposition 4.4 and Theorem 4.5 involving the pseudo-Riemannian manifold $(\mathbb{R}^D, f^*H)$ and the quotient group $(\mathcal{P}, d_{\mathcal{P}})$ with Euclidean-induced metric.

To ease the readability, we recall the two involved notions of distance and their respective definition:

- $d_{f^*H}(\mathbf{w}_0, \mathbf{w}_1)$ the geodesic distance for any two parameters $\mathbf{w}_0, \mathbf{w}_1 \in \mathbb{R}^D$

- $d_{\mathcal{P}}([\mathbf{w}_0, \mathbf{w}_1])$ the quotient Euclidean distance for any two equivalence classes $[\mathbf{w}_0], [\mathbf{w}_1] \in \mathcal{P}$

$$d_{f^*H}(\mathbf{w}_0, \mathbf{w}_1) = \inf_\gamma \text{LEN}_{f^*H}(\gamma) \tag{17}$$

$$= \inf_\gamma \int_0^1 \|\gamma'(t)\|_{f^*H_{\gamma(t)}} \mathrm{d}t \tag{18}$$

$$= \inf_\gamma \int_0^1 \sqrt{\gamma'(t)^\top \cdot f^*H_{\gamma(t)} \cdot \gamma'(t)} \mathrm{d}t, \tag{19}$$

where the infimum is taken over smoothly differentiable curves $\gamma : [0,1] \to \mathbb{R}^D$ such that $\gamma(0) = \mathbf{w}_0$ and $\gamma(1) = \mathbf{w}_1$.

$$d_{\mathcal{P}}([\mathbf{w}_0], [\mathbf{w}_1]) = \inf \left\{ \sum_{i=1}^n \|p_i - q_i\| \right\}, \tag{20}$$

where the infimum is taken over all finite sequences $\{p_i\}_{i=1\dots n}, \{q_i\}_{i=1\dots n} \subset \mathbb{R}^D$ such that $[\mathbf{w}_0] = [p_1], [p_{i+1}] = [q_i]$ and $[q_n] = [\mathbf{w}_1]$.

Let us state a theorem that encapsulate together both the 0-distance part in Proposition 4.4 and the $\epsilon$-$\delta$ part in Theorem 4.5 in a more unified way.

**Theorem B.1.** *For any $\mathbf{w}_0, \mathbf{w}_1 \in \mathbb{R}^D$ it holds*

$$d_{f^*H}(\mathbf{w}_0, \mathbf{w}_1) = 0 \iff d_{\mathcal{P}}([\mathbf{w}_0], [\mathbf{w}_1]) = 0, \tag{21}$$

*and also that, for any $\epsilon > 0$ there exists $\delta > 0$ such that*

$$d_{\mathcal{P}}([\mathbf{w}_0], [\mathbf{w}_1]) < \delta \implies d_{f^*H}(\mathbf{w}_0, \mathbf{w}_1) < \epsilon \tag{22}$$

$$d_{f^*H}(\mathbf{w}_0, \mathbf{w}_1) < \delta \implies d_{\mathcal{P}}([\mathbf{w}_0], [\mathbf{w}_1]) < \epsilon, \tag{23}$$

We prove the 3 points separately, in Appendix B.1, Appendix B.2 and Appendix B.3 respectively.

## B.1  Proof of Eq. 21

The proof logic is

$$d_{f^*H}(\mathbf{w}_0, \mathbf{w}_1) = 0 \iff [\mathbf{w}_0] = [\mathbf{w}_1] \iff d_{\mathcal{P}}([\mathbf{w}_0], [\mathbf{w}_1]) = 0. \tag{24}$$

and we prove the two steps in the two following Propositions, respectively.

**Proposition B.2.** *For any $\mathbf{w}_0, \mathbf{w}_1 \in \mathbb{R}^D$ it holds*

$$d_{f^*H}(\mathbf{w}_0, \mathbf{w}_1) = 0 \iff [\mathbf{w}_0] = [\mathbf{w}_1]. \tag{25}$$

*Proof.* By definition $[\mathbf{w}_0] = [\mathbf{w}_1]$ if and only if there exists a piecewise differentiable $\gamma : [0,1] \to \mathbb{R}^D$ such that $\gamma(0) = \mathbf{w}_0, \gamma(1) = \mathbf{w}_1$ and $f(\mathbf{w}_0, \mathbf{x}) = f(\gamma(t), \mathbf{x})$ for any $t \in [0,1]$ and $\mathbf{x} \in \mathcal{X}$. Then
$\boxed{\Longleftarrow}$ Consider a $\gamma$ from the definition of the equivalence relation $\sim$ and define the points $\mathbf{w}_t = \gamma(t)$ for ease of notation. Then for any $t \in [0,1]$

$$\gamma'(t) = \lim_{\epsilon \to 0} \frac{\mathbf{w}_{t+\epsilon} - \mathbf{w}_t}{\epsilon}. \tag{26}$$

For any $\mathbf{x} \in \mathcal{X}$ it holds that $f(\mathbf{w}_t, \mathbf{x}) = f(\mathbf{w}_{t'}, \mathbf{x}) \forall t, t' \in [0,1]$, which implies that $f(\mathbf{w}_{t+\epsilon}, \mathbf{x}) - f(\mathbf{w}_t, \mathbf{x}) = 0 \ \forall t \in [0,1] \forall \epsilon \in [0, 1-t]$. Thus,

$$0 = \lim_{\epsilon \to 0} \frac{f(\mathbf{w}_{t+\epsilon}, \mathbf{x}) - f(\mathbf{w}_t, \mathbf{x})}{\epsilon} = \mathbf{J}_{\mathbf{w}_t}(\mathbf{x}) \cdot \lim_{\epsilon \to 0} \frac{\mathbf{w}_{t+\epsilon} - \mathbf{w}_t}{\epsilon} = \mathbf{J}_{\mathbf{w}_t}(\mathbf{x}) \cdot \gamma'(t). \tag{27}$$

This holds to any $\mathbf{x} \in \mathcal{X}$, so the same holds for the per-datum stacked jacobians $\mathbf{J}_{\mathbf{w}_t} \cdot \gamma'(t) = 0$. Thus,

$$\|\gamma'(t)\|_{f^*H_{\gamma(t)}}^2 = \gamma'(t)^\top \cdot f^*H_{\gamma(t)} \cdot \gamma'(t) = \gamma'(t)^\top \cdot \mathbf{J}_{\mathbf{w}_t}^\top \mathbf{H}_{\mathbf{w}_t} \mathbf{J}_{\mathbf{w}_t} \cdot \gamma'(t) = 0 \tag{28}$$

and we can measure the length of $\gamma$ in the pullback metric as

$$\text{LEN}_{f^*H}(\gamma) = \int_0^1 \|\gamma'(t)\|_{f^*H_{\gamma(t)}} \mathrm{d}t = \int_0^1 0 \, \mathrm{d}t = 0, \tag{29}$$

which gives an upper bound on the geodesic distance

$$d_{f^*H}(\mathbf{w}_0, \mathbf{w}_1) = \inf_{\hat{\gamma}} \text{LEN}_{f^*H}(\hat{\gamma}) \leq \text{LEN}_{f^*H}(\gamma) = 0, \tag{30}$$

thus, $d_{f^*H}(\mathbf{w}_0, \mathbf{w}_1) = 0$ and this implication is proven.

$\boxed{\implies}$ $d_{f^*H}(\mathbf{w}_0, \mathbf{w}_1) = 0$ implies that there exists a 0-length differentiable $\gamma : [0, 1] \to \mathbb{R}^D$ such that $\gamma(0) = \mathbf{w}_0, \gamma(1) = \mathbf{w}_1$. Without loss of generality, we can assume $\gamma$ to be non-stationary, i.e. $\gamma'(t) \neq 0$. Here 0-length means

$$0 = \text{LEN}_{f^*H}(\gamma) = \int_0^1 \|\gamma'(t)\|_{f^*H_{\gamma(t)}} \mathrm{d}t, \tag{31}$$

which implies that $\|\gamma'(t)\|_{f^*H_{\gamma(t)}} = 0$ for any $t \in [0, 1]$ except a zero-measure set which we can neglect later. Then

$$0 = \|\gamma'(t)\|_{f^*H_{\gamma(t)}}^2 = \gamma'(t)^\top \cdot \mathbf{J}_{\mathbf{w}_t}^\top \mathbf{H}_{\mathbf{w}_t} \mathbf{J}_{\mathbf{w}_t} \cdot \gamma'(t) \quad \implies \quad \mathbf{J}_{\mathbf{w}_t} \cdot \gamma'(t) = 0, \tag{32}$$

by positive definitess of $\mathbf{H}_{\mathbf{w}_t}$, assumed as hypothesis. We highlight that the leftmost 0 in the previous equation is a scalar, while the rightmost 0 is a vector in $\mathbb{R}^{NO}$ as obtained by the matrix-vector product of $\mathbf{J}_{\mathbf{w}_t} \in \mathbb{R}^{NO \times D}$ with $\gamma'(t) \in \mathbb{R}^D$. Looking at the equation $\mathbf{J}_{\mathbf{w}_t} \cdot \gamma'(t) = 0$ componentwise implies that

$$\langle \nabla_{\mathbf{w}} [f(\mathbf{w}_t, \mathbf{x})]_o, \gamma'(t) \rangle = 0 \quad \forall \mathbf{x} \in \mathcal{X}, \forall o \in \{1, \dots, O\}, \forall t \in [0, 1], \tag{33}$$

where $[v]_o$ refers to the $o$th component of a vector $v$. Thus, for $T \in [0, 1]$, by Fundamental Theorem of Calculus we have

$$[f(\mathbf{w}_T, \mathbf{x})]_o - [f(\mathbf{w}_0, \mathbf{x})]_o = \int_0^T \langle \nabla_{\mathbf{w}} [f(\mathbf{w}_t, \mathbf{x})]_o, \gamma'(t) \rangle \mathrm{d}t = 0 \qquad \forall \mathbf{x} \in \mathcal{X}, \forall o \in \{1, \dots, O\}. \tag{34}$$

Then $f(\mathbf{w}_T, \mathbf{x}) = f(\mathbf{w}_0, \mathbf{x}) \, \forall \mathbf{x} \in \mathcal{X}$ and $\forall T \in [0, 1]$. So we proved that $\gamma$ is an homotopy of $(\mathbf{w}, \mathbf{w}')$ such that $\gamma(t) \in \mathcal{R}_{\mathcal{X}}^f(\mathbf{w}) \, \forall t \in [0, 1]$. Thus,

$$\mathbf{w}' \in \bar{\mathcal{R}}_{\mathcal{X}}^f(\mathbf{w}) \quad \implies \quad \mathbf{w}' \sim \mathbf{w} \quad \implies \quad [\mathbf{w}'] = [\mathbf{w}], \tag{35}$$

and this completes the proof. $\qquad \square$

**Proposition B.3.** *For any $\mathbf{w}_0, \mathbf{w}_1 \in \mathbb{R}^D$ it holds*

$$[\mathbf{w}_0] = [\mathbf{w}_1] \quad \iff \quad d_{\mathcal{P}}([\mathbf{w}_0, \mathbf{w}_1]) = 0. \tag{36}$$

*Proof.* $\boxed{\implies}$ This arrow is trivially true by considering the two sequences in the definition of the quotient distance to be of length one and such that $p_1 = \mathbf{w}_0$ and $q_1 = \mathbf{w}_1$.

$\boxed{\impliedby}$ $d_{\mathcal{P}}([\mathbf{w}_0, \mathbf{w}_1]) = 0$ implies that, by definition of inf, there exists a sequence $\epsilon_m \to 0$ such that $\forall m \in \mathbb{N}$ there exists two finite sequences of points $p_1^{(m)}, \dots, p_n^{(m)} \in \mathbb{R}^D$ and $q_1^{(m)}, \dots, q_n^{(m)} \in \mathbb{R}^D$ such that

$$\sum_{i=1}^n \|p_i^{(m)} - q_i^{(m)}\| = \epsilon_m, \tag{37}$$

where $[\mathbf{w}_0] = [p_1^{(m)}]$, $[p_{i+1}^{(m)}] = [q_i^{(m)}]$ and $[q_n^{(m)}] = [\mathbf{w}_1]$. Note that the sequence length $n$ may depend on $m$.

Lipschitzness of $f$ in parameters, assumed by hypothesis, means that

$$\|p - q\| < \epsilon \quad \implies \quad \|f(p, \mathbf{x}) - f(q, \mathbf{x})\| < L\epsilon \qquad \forall p, q \in \mathbb{R}^D, \forall \mathbf{x} \in \mathcal{X}, \forall \epsilon > 0, \tag{38}$$

thus,

$$\sum_{i=1}^n \|p_i^{(m)} - q_i^{(m)}\| = \epsilon_m \quad \implies \quad \sum_{i=1}^n \|f(p_i^{(m)}, \mathbf{x}) - f(q_i^{(m)}, \mathbf{x})\| < 2L\epsilon_m \qquad \forall \mathbf{x} \in \mathcal{X}. \tag{39}$$

Also, by definition of the equivalence class, it holds $\forall \mathbf{x} \in \mathcal{X}$ that

$$[\mathbf{w}_0] = [p_1^{(m)}] \quad \implies \quad \|f(\mathbf{w}_0, \mathbf{x}) - f(p_1^{(m)}, \mathbf{x})\| = 0 \tag{40}$$

$$[p_{i+1}^{(m)}] = [q_i^{(m)}] \quad \implies \quad \|f(p_{i+1}^{(m)}, \mathbf{x}) - f(q_i^{(m)}, \mathbf{x})\| = 0 \tag{41}$$

$$[q_n^{(m)}] = [\mathbf{w}_1] \quad \implies \quad \|f(q_n^{(m)}, \mathbf{x}) - f(\mathbf{w}_1, \mathbf{x})\| = 0. \tag{42}$$

Thus, by the triangular inequality, the last four equations imply

$$\|f(\mathbf{w}_0, \mathbf{x}) - f(\mathbf{w}_1, \mathbf{x})\| \leq \|f(\mathbf{w}_0, \mathbf{x}) - f(p_1^{(m)}, \mathbf{x})\| + \|f(p_1^{(m)}, \mathbf{x}) - f(q_n^{(m)}, \mathbf{x})\| \tag{43}$$
$$+ \|f(q_n^{(m)}, \mathbf{x}) - f(\mathbf{w}_1, \mathbf{x})\|$$

$$= \|f(p_1^{(m)}, \mathbf{x}) - f(q_n^{(m)}, \mathbf{x})\| \tag{44}$$

$$\leq \sum_{i=1}^{n} \|f(p_i^{(m)}, \mathbf{x}) - f(q_i^{(m)}, \mathbf{x})\| + \sum_{i=1}^{n-1} \|f(p_{i+1}^{(m)}, \mathbf{x}) - f(q_i^{(m)}, \mathbf{x})\| \tag{45}$$

$$< 2L\epsilon_m + \sum_{i=1}^{n-1} 0 = 2L\epsilon_m, \tag{46}$$

and this holds for any $m \in \mathbb{N}$. Taking the limit $m \to \infty$, $\epsilon_m \to 0$ implies $\|f(\mathbf{w}_0, \mathbf{x}) - f(\mathbf{w}_1, \mathbf{x})\| \leq 0$. Thus, $f(\mathbf{w}_0, \mathbf{x}) = f(\mathbf{w}_1, \mathbf{x}) \ \forall \mathbf{x} \in \mathcal{X}$, and, thus, $[\mathbf{w}_0] = [\mathbf{w}_1]$, which completes the proof. $\qquad\square$

## B.2 Proof of Eq. 22

In order to prove Eq. 22, we first prove a weaker statement

**Proposition B.4.** *For any $\mathbf{w}_0, \mathbf{w}_1 \in \mathbb{R}^D$ and for any $\delta > 0$ it holds that*

$$\|\mathbf{w}_0 - \mathbf{w}_1\| < \delta \quad \implies \quad d_{f^*H}(\mathbf{w}_0, \mathbf{w}_1) < L\delta. \tag{47}$$

*Proof.* Let $\gamma : [0, 1] \to \mathbb{R}^D$ be defined as $\gamma(t) = (1-t)\mathbf{w}_0 + t\mathbf{w}_1$, then

$$\text{LEN}_{f^*H}(\gamma) = \int_0^1 \|\gamma'(t)\|_{f^*H_{\gamma(t)}} \mathrm{d}t = \int_0^1 \|\mathbf{w}_1 - \mathbf{w}_0\|_{f^*H_{\gamma(t)}} \mathrm{d}t$$
$$\leq \int_0^1 L\|\mathbf{w}_1 - \mathbf{w}_0\| \mathrm{d}t = L\|\mathbf{w}_1 - \mathbf{w}_0\|, \tag{48}$$

and thus,

$$d_{f^*H}(\mathbf{w}_0, \mathbf{w}_1) = \inf_{\hat{\gamma}} \text{LEN}_{f^*H}(\hat{\gamma}) \leq \text{LEN}_{f^*H}(\gamma) = L\|\mathbf{w}_1 - \mathbf{w}_0\| < L\delta. \tag{49}$$

$$\square$$

**Definition B.5.** Let $\gamma_1, \ldots, \gamma_n : [0, 1] \to \mathbb{R}^D$ a sequence of piecewise differentiable paths such that

$$\gamma_{i-1}(1) = \gamma_i(0) \quad \forall i \in \{0, \ldots, n\}. \tag{50}$$

Consider the piecewise differentiable path that is the *concatenation* of the paths one after the other, $\hat{\gamma} = \text{CAT}(\gamma_1, \ldots, \gamma_n)$ defined as

$$\hat{\gamma}(t) = \gamma_i(nt - i) \quad \text{if } i \leq t \leq i+1. \tag{51}$$

It is straightforward to see that $\text{LEN}(\hat{\gamma}) = \sum_{i=1}^{n} \text{LEN}(\gamma_i)$.

The choice of $\epsilon, \delta$ that prove Eq. 22 trivially follows from the following

**Proposition B.6.** *With the assumption of eigenvalue upper bound $L$, for any $\mathbf{w}_0, \mathbf{w}_1 \in \mathbb{R}^D$ for any $\delta > 0$ it holds that*

$$d_{\mathcal{P}}([\mathbf{w}_0, \mathbf{w}_1]) < \delta \quad \implies \quad d_{f^*H}(\mathbf{w}_0, \mathbf{w}_1) < 3L\delta. \tag{52}$$

*Proof.* $d_{\mathcal{P}}([\mathbf{w}_0, \mathbf{w}_1]) < \delta$ implies that, by the definition of inf, there exists two sequences $\{p_i\}_{i=1\ldots n}, \{q_i\}_{i=1\ldots n} \subset \mathbb{R}^D$ such that $[\mathbf{w}_0] = [p_1]$, $[p_{i+1}] = [q_i]$, $[q_n] = [\mathbf{w}_1]$ and such that

$$\sum_{i=1}^{n} \|p_i - q_i\| < 2\delta. \tag{53}$$

Now the idea is to define $n+1$ paths on the equivalence classes and $n$ paths connecting them, then the stacking of the $2n+1$ will give an upper bound on the geodesic distance.

Let us first define the paths $\gamma_{2i}$ for $i = 0, \ldots, n$ by making use of Proposition B.2

- $[\mathbf{w}_0] = [p_1]$ imply that there exists $\gamma_0 : [0,1] \to \mathbb{R}^D$ be such that $\gamma_0(0) = \mathbf{w}_0$, $\gamma_0(1) = p_1$ and such that $\mathrm{LEN}_{f^*H}(\gamma_0) < L\delta/n+1$

- $[q_n] = [\mathbf{w}_1]$ imply that there exists $\gamma_{2n} : [0,1] \to \mathbb{R}^D$ be such that $\gamma_{2n}(0) = q_n$, $\gamma_{2n}(1) = \mathbf{w}_1$ and such that $\mathrm{LEN}_{f^*H}(\gamma_{2n}) < L\delta/n+1$

- for all $i = 1, \ldots, n-1$, $[p_{i+1}] = [q_i]$ imply that there exists $\gamma_{2i} : [0,1] \to \mathbb{R}^D$ be such that $\gamma_{2i}(0) = p_{i+1}$, $\gamma_{2i}(1) = q_i$ and such that $\mathrm{LEN}_{f^*H}(\gamma_{2i}) < L\delta/n+1$

And then for all $i = 1, \ldots, n$, Proposition B.4 and $\sum_{i=1}^n \|p_i - q_i\| < 2\delta$ imply $\gamma_{2i-1} : [0,1] \to \mathbb{R}^D$ be such that $\gamma_{2i-1}(0) = p_i$, $\gamma_{2i-1}(1) = q_i$ and such that $\sum_{i=1}^n \mathrm{LEN}_{f^*H}(\gamma_{2i-1}) < 2L\delta$.

Then concatenating these $2n+1$ paths, we have

$$d_{f^*H}(\mathbf{w}_0, \mathbf{w}_1) = \inf_{\hat{\gamma}} \mathrm{LEN}_{f^*H}(\hat{\gamma}) \leq \mathrm{LEN}_{f^*H}(\mathrm{CAT}(\gamma_0, \ldots, \gamma_{2n})) \tag{54}$$

$$= \sum_{i=0}^{2n} \mathrm{LEN}_{f^*H}(\gamma_i) = \sum_{i=1}^n \mathrm{LEN}_{f^*H}(\gamma_{2i-1}) + \sum_{i=0}^n \mathrm{LEN}_{f^*H}(\gamma_{2i}) < 2L\delta + L\delta. \tag{55}$$

$\square$

## B.3 Proof of Eq. 23

**Proposition B.7.** *The pullback metric does not "rotate" too much when moving from $\mathbf{w}$ to $\mathbf{w} + \epsilon$, formally*

$$(1 - K\|\epsilon\|)\|v\|_{f^*H_{\mathbf{w}+\epsilon}} \leq \|v\|_{f^*H_{\mathbf{w}}} \leq (1 + K\|\epsilon\|)\|v\|_{f^*H_{\mathbf{w}+\epsilon}}. \tag{56}$$

*Proof.* Follows from the $K$-Lischitz assumption on $\mathbf{w} \mapsto \mathbf{J}_{\mathbf{w}}$ $\square$

The choice of $\epsilon, \delta$ that prove Eq. 23 trivially follows from the following

**Proposition B.8.** *With the assumption of eigenvalue upper bound $L$, for any $\mathbf{w}_0, \mathbf{w}_1 \in \mathbb{R}^D$ for any $\delta > 0$ it holds that*

$$d_{f^*H}(\mathbf{w}_0, \mathbf{w}_1) < \delta \quad \implies \quad d_{\mathcal{P}}([\mathbf{w}_0, \mathbf{w}_1]) < \frac{6\delta}{l}. \tag{57}$$

*Proof.* $d_{f^*H}(\mathbf{w}_0, \mathbf{w}_1) < \delta$ implies that there exist a path $\gamma : [0,1] \to \mathbb{R}^D$ with

$$2\delta = \mathrm{LEN}_{f^*H}(\gamma) = \int_0^1 \|\gamma'(t)\|_{f^*H_{\gamma(t)}} \mathrm{d}t. \tag{58}$$

Consider also the Euclidean length $\mathrm{LEN}_E(\gamma)$ which does *not* depend on $\delta$. Then, for any $\delta$ there exist $n(\delta) \in \mathbb{N}$ such that, considering the uniform partition of $[0,1]$, $t_i = i/n(\delta)$ for $i = 0, \ldots, n(\delta)$ it holds the discrete approximation of the integral

$$3\delta > \sum_{i=0}^{n(\delta)-1} \|\gamma(t_{i+1}) - \gamma(t_i)\|_{f^*H_{\gamma(t_i)}} = \sum_{i=0}^{n(\delta)-1} \|\gamma(t_{i+1}) - p_i + p_i - \gamma(t_i)\|_{f^*H_{\gamma(t_i)}} \tag{59}$$

$$= \sum_{i=0}^{n(\delta)-1} \|\gamma(t_{i+1}) - p_i\|_{f^*H_{\gamma(t_i)}} + \underbrace{\|p_i - \gamma(t_i)\|_{f^*H_{\gamma(t_i)}}}_{=0}, \tag{60}$$

where $p_i$ is defined for every $i \in \{0, \ldots, n(\delta) - 1\}$ as follow. Consider the projection $P_i^K$ on the kernel of $f^*H_{\gamma(t_i)}$, and the projection $P_i^I$ on the image of $f^*H_{\gamma(t_i)}$, such that the two projections are orthogonal and $P_i^K + P_i^I = \mathbf{I}$. Define the point $p_i \in \mathbb{R}^D$ as $p_i = P_i^K(\gamma(t_{i+1}) - \gamma(t_i)) + \gamma(t_i)$, which implies that $\|p_i - \gamma(t_i)\|_{f^*H_{\gamma(t_i)}} = 0$. By definition of the projections, it also hold that $p_i = \gamma(t_{i+1}) - P_i^I(\gamma(t_{i+1}) - \gamma(t_i))$, which implies that $\gamma(t_{i+1}) - p_i = P_i^I(\gamma(t_{i+1}) - \gamma(t_i))$ is

aligned with the non-zero eigenvalues of $f^* H_{\gamma(t_i)}$, this means that we can resort to the lower bound $l$ on the non-zero eigenvalues and Proposition B.7 to see that

$$\|\gamma(t_{i+1}) - p_i\|_{f^* H_{\gamma(t_i)}} \geq \underbrace{(1 - K\|\gamma(t_{i+1}) - \gamma(t_i)\|)}_{\tilde{K}}\|\gamma(t_{i+1}) - p_i\|_{f^* H_{\gamma(t_{i+1})}} \qquad (61)$$

$$\geq \tilde{K}l\|\gamma(t_{i+1}) - p_i\| \qquad (62)$$

$$\geq \frac{l}{2}\|\gamma(t_{i+1}) - p_i\| \qquad \forall i \in \{0, \ldots, n(\delta) - 1\}, \qquad (63)$$

where, recalling that $\|\gamma(t_{i+1}) - \gamma(t_i)\| = 1/n(\delta)$ by construction, there always exist an $n(\delta)$ big enough such that $\tilde{K} \geq 1/2$, which we can assume without loss of generality. Rearranging the terms, we have

$$\frac{6\delta}{l} \geq \sum_{i=0}^{n(\delta)-1} \|\gamma(t_{i+1}) - p_i\|. \qquad (64)$$

Finally, we can define the points $q_i = \gamma(t_{i+1})$ and we have the two sequences $p_0, \ldots, p_{n(\delta)-1}$ and $q_0, \ldots, q_{n(\delta)-1}$ whose satisfies the costrains $[\mathbf{w}_0] = [p_0]$, $[p_{i+1}] = [q_i]$ and $[q_{n(\delta)-1}] = [\mathbf{w}_1]$. We highlight that the dependence on $\delta$ of the length of the sequence is not problematic, as it is sufficient that $n(\delta)$ is finite for every fixed $\delta$, which it is. Thus,

$$d_{\mathcal{P}}([\mathbf{w}_0, \mathbf{w}_1]) \leq \sum_{i=0}^{n(\delta)-1} \|p_i - q_i\| < \frac{6\delta}{l}, \qquad (65)$$

which concludes the proof. $\qquad\qquad\square$

## C   Proof of existence of the two Riemannian manifolds

This section contains the proof for the existence of the two Riemannian manifolds $(\mathcal{P}_{\bar{\mathbf{w}}}, \mathfrak{m})$ and $(\mathcal{P}_{\bar{\mathbf{w}}}^\perp, \mathfrak{m}^\perp)$ embedded in $\mathbb{R}^D$. In the rest of the section, we consider a fixed $\bar{\mathbf{w}} \in \mathbb{R}^D$.

Fix a training set $\mathcal{X} = \{\mathbf{x}_1, \ldots, \mathbf{x}_N\} \subset \mathbb{R}^I$ of size $N$ and consider the stacked partial evaluation of the network defined as

$$\mathfrak{F} : \mathbb{R}^D \longrightarrow \mathbb{R}^{NO} \qquad (66)$$

$$\mathbf{w} \longmapsto (f(\mathbf{w}, \mathbf{x}_1), \ldots, f(\mathbf{w}, \mathbf{x}_N)). \qquad (67)$$

And define $\bar{\mathbf{y}} = \mathfrak{F}(\bar{\mathbf{w}}) \in \mathbb{R}^{NO}$. The differential $\nabla_{\mathbf{w}}\mathfrak{F}\big|_{\bar{\mathbf{w}}} = \mathbf{J}_{\bar{\mathbf{w}}} \in \mathbb{R}^{NO \times D}$ equals the stacking of the per-datum Jacobians, and we assume it to be full rank thanks to the uniform lower bound on the eigenvalues of the NTK matrix. Full-rankness in the overparametrized setting $D > NO$ implies the surjectivity of the differential operator.

$$\mathbf{J}_{\bar{\mathbf{w}}} = \left(\frac{\partial \mathfrak{F}_i}{\partial \mathbf{w}_j}\bigg|_{\bar{\mathbf{w}}}\right)_{i=1,\ldots,NO; j=1,\ldots,D}. \qquad (68)$$

We reorder the $\mathbf{w}_i$ so that the first $NO$ columns are independent. Then the $NO \times NO$ matrix

$$R = \left(\frac{\partial \mathfrak{F}_i}{\partial \mathbf{w}_j}\bigg|_{\bar{\mathbf{w}}}\right)_{i=1,\ldots,NO; j=1,\ldots,NO} \qquad (69)$$

is non-singular. We consider the map

$$\alpha(\mathbf{w}_1, \ldots, \mathbf{w}_D) = (\mathfrak{F}(\mathbf{w})_1, \ldots, \mathfrak{F}(\mathbf{w})_{NO}, \mathbf{w}_{NO+1}, \ldots, \mathbf{w}_D). \qquad (70)$$

We obtain

$$\nabla_{\mathbf{w}}\alpha\big|_{\bar{\mathbf{w}}} = \left(\frac{\partial \alpha_i}{\partial \mathbf{w}_j}\bigg|_{\bar{\mathbf{w}}}\right)_{i=1,\ldots,D; j=1,\ldots,D} = \begin{pmatrix} R & * \\ 0 & \mathbb{I} \end{pmatrix}, \qquad (71)$$

and this is non-singular. By the inverse function theorem, $\alpha$ is a local diffeomorphism. So there is an open $W \subseteq \mathbb{R}^D$ containing $\bar{w}$ such that $\alpha|_W : W \to \alpha(W)$ is smooth with smooth inverse.

Finally, define

$$\mathcal{P}_{\bar{\mathbf{w}}}^{\perp} = \left\{ \alpha^{-1}(\underbrace{\bar{\mathbf{y}}_1, \ldots, \bar{\mathbf{y}}_{NO}}_{NO}, p_1, \ldots, p_{D-NO}) \quad \text{for } p \in \mathbb{R}^{D-NO} \right\} \subseteq \mathbb{R}^D, \qquad (72)$$

and similarly

$$\mathcal{P}_{\bar{\mathbf{w}}} = \left\{ \alpha^{-1}(p_1, \ldots, p_{NO}, \underbrace{0, \ldots, 0}_{D-NO}) \quad \text{for } p \in \mathbb{R}^{NO} \right\} \subseteq \mathbb{R}^D. \qquad (73)$$

We claim that the two restrictions of $\alpha$ are slice charts of $\mathcal{P}_{\bar{\mathbf{w}}}^{\perp}$ and $\mathcal{P}_{\bar{\mathbf{w}}}$, respectively. Since it is a smooth diffeomorphism, it is certainly a chart. Moreover, by construction, the points in $\mathcal{P}_{\bar{\mathbf{w}}}^{\perp}$ are exaclty those whose image under $\mathfrak{F}$ is $\bar{y}$, thus, $\mathcal{P}_{\bar{\mathbf{w}}}^{\perp} \subseteq [\bar{\mathbf{w}}]$. On the other hand, the points $\mathbf{w} \in \mathcal{P}_{\bar{\mathbf{w}}}$ parametrize functions that take the values $\mathfrak{F}(\mathbf{w}) = p$ in a local neighbourhood of $\bar{y}$. Thus, locally it never intersects the same equivalence class more than one time.

# D   Proof of Theorem 5.1

By definition of the kernel manifold, we have that if $\mathbf{w} \in \mathcal{P}_{\mathbf{w}'}^{\perp}$, then we have that $\mathbf{w} \in [\mathbf{w}']$. Hence for all $\mathbf{w} \in \mathcal{P}_{\mathbf{w}'}^{\perp}$ and for all $\mathbf{x} \in \mathcal{X}$

$$f(\mathbf{w}, \mathbf{x}) = f(\mathbf{w}', \mathbf{x}). \qquad (74)$$

It follows that $\text{Var}_{\mathbf{w} \sim \mathcal{P}_{\mathbf{w}'}^{\perp}} [f(\mathbf{w}, \mathbf{x})] = 0$ for any $\mathbf{x} \in \mathcal{X}$.

For the second statement notice that $\text{Var}_{\mathbf{w} \sim \mathcal{P}_{\mathbf{w}'}^{\perp}} [f(\mathbf{w}, \mathbf{x}_{test}] = 0$ if and only if $f(\mathbf{w}, \mathbf{x}_{test}) = c$ for all $\mathbf{w} \sim \mathcal{P}_{\mathbf{w}'}^{\perp}$ and for some constant $c$.

Suppose $\hat{\mathbf{w}} \in \bar{\mathcal{R}}_{\mathcal{X}}^{f}$ and $\hat{\mathbf{w}} \notin \bar{\mathcal{R}}_{\mathcal{X} \cup \{\mathbf{x}_{test}\}}^{f}$, then $f(\hat{\mathbf{w}}, \mathbf{x}_{test}) \neq f(\mathbf{w}', \mathbf{x}_{test})$, which means that $f(\mathbf{w}, \mathbf{x}_{test})$ is not constant. Hence we have that $\text{Var}_{\mathbf{w} \sim \mathcal{P}_{\mathbf{w}'}^{\perp}} [f(\mathbf{w}, \mathbf{x}_{test}] > 0$

# E   Further results and experimental setup

## E.1   Implementation details of the Laplace approximation

Sampling from Laplace's approximation requires computing the inverse square root of a matrix of size $D \times D$, where $D$ is the number of parameters. For most models this problem is intractable. The standard approach to this problem is to consider sparse approximations to the Hessian of the loss function such as KFAC, Last-Layer, and Diagonal approximations. However, these approximations introduce additional complexity making the task of validating our theoretical analysis much harder. In light of these considerations, we choose to sample from Laplace's approximation in a way that is closest to the theoretical ideal, at the cost of performing expensive computations.

In small experiments with the toy regression problem, we instantiate the exact GGN and compute vector products with its inverse-square root. For experiments with LeNet and ResNet, we rely on the empirical observation that the spectrum of the GGN is dominated by its leading eigenvalues (Figure 6)

This makes low-rank approximations of the GGN particularly attractive. We choose the Lanczos algorithm (Lanczos, 1950) with full reorthogonalization and run it for a very high number of iterations, to ensure numerical stability and very low reconstruction error to form our low-rank approximations of the GGN. Additionally, Lanczos only requires implicit access to the matrix so we avoid the memory cost and GGN vector products for neural networks can be performed efficiently using Jacobian-Vector products and Vector-Jacobian products. If we do

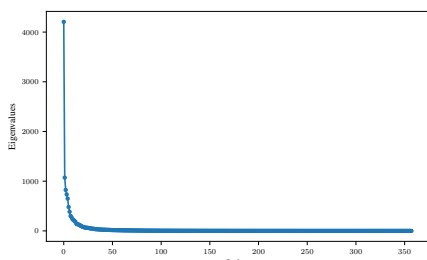

Figure 6: Eigenvalues of the GGN of a Convolutional Neural Network trained on MNIST.

---

**Algorithm 1** Laplace diffusion

---

1: **Input:** Observed data $\mathcal{D}$, trained MAP point $\mathbf{w}'$, number of steps $T$, number of samples $S$, rank $k$.
2: Initialize samples $\mathbf{w}_1^0 \ldots \mathbf{w}_S^0$ as the MAP estimate $\mathbf{w}'$
3: **for** $j$ **in** $1, \ldots S$ **do**
4:     **for** $t$ **in** $1, \ldots T$ **do**
5:         Sample $\epsilon \sim \mathcal{N}(0, \mathbf{I}_k)$
6:         Compute the top-$k$ eigenvalues($\Lambda_j^t$) and eigenvectors($U_j^t$) of GGN$_{\mathbf{w}_j^t}$
7:         $\mathbf{w}_j^t \leftarrow \mathbf{w}_j^t + \frac{1}{\sqrt{N}} U_j^t (\Lambda_j^t + \alpha \mathbf{I})^{-\frac{1}{2}} \epsilon$
8:     **end for**
9: **end for**
10: Return posterior samples $\mathbf{w}_1^T \ldots \mathbf{w}_S^T$.

---

a sufficiently large number of iterations we obtain the non-zero eigenvalues $\tilde{\mathbf{\Lambda}}$, and corresponding eigenvectors $\mathbf{U_1}$. For obtaining samples from the diffusion we can use Algorithm 1 with the eigenvalues and eigenvectors computed using the Lanczos algorithm.

Given the non-zero eigenvalues $\tilde{\mathbf{\Lambda}}$, and corresponding eigenvectors $\mathbf{U_1}$ we can also form inverse vector products with the square root of GGN $+ \alpha I$. It should be evident from the discussion in section 3 about decomposing the covariance that this vector product with a vector $v$ is given by:

$$(\text{GGN} + \alpha I)^{-\frac{1}{2}} v = \mathbf{U_1}(\tilde{\mathbf{\Lambda}} + \alpha \mathbf{I}_k)^{-\frac{1}{2}} v + \frac{1}{\sqrt{\alpha}} \mathbf{U_2} v \tag{75}$$

$$= \mathbf{U_1}(\tilde{\mathbf{\Lambda}} + \alpha \mathbf{I}_k)^{-\frac{1}{2}} v + \frac{1}{\sqrt{\alpha}} (\mathbf{I} - \mathbf{U_1}) v \tag{76}$$

$$= \mathbf{U_1}((\tilde{\mathbf{\Lambda}} + \alpha \mathbf{I}_k)^{-\frac{1}{2}} - \frac{1}{\sqrt{\alpha}} \mathbf{I}_k) v + \frac{1}{\sqrt{\alpha}} v \tag{77}$$

This allows us to form inverse-square root vector products with the GGN $+ \alpha I$ given the non-zero eigenspectrum. We run the sampling algorithm on H100 GPUs to run the high-order Lanczos decomposition. This approach of sampling from Laplace's approximation has $O(pk^2)$ time complexity, and $O(pk)$ memory cost, where $k$ is the number of Lanczos iterations and $p$ is the number of parameters.

### E.2 Experimental details and further results for toy experiments

#### E.2.1 Toy regression in Figure 2

In this experiment, we fit a small MLP, with 2 hidden layers of width 10 on the sine curve. Due to the small size of the GGN it is possible to instantiate and do all the computations explicitly. We sample from the exact Laplace's approximation, the non-kernel and kernel subspace of the GGN, and use the neural network and the linearized predictive functions for the top row and the bottom row respectively. For the middle, we simulate a diffusion on the kernel manifold for the center plot, a diffusion in the non-kernel manifold for the right plot and we do alternating steps in the two manifolds for the left plot which gives us the full distribution.

We also do a similar experiment to show that the same phenomenon also holds for classification. We use a small convolutional neural network, with 2 convolutional layers with kernel of size 3, to classify a 2-class mixture of Gaussians and look at uncertainties of sampled Laplace, linearized Laplace, and Laplace's diffusion. We decompose these uncertainties into their kernel and non-kernel components respectively. We see the same effect for classification as we did in regression in Figure 7

The main takeaway of these experiments is that sampled Laplace underfits in-distribution and this effect is related to the kernel component of the distribution.

#### E.2.2 Effect of kernel rank on in-distribution fit in figure 4

For this experiment, we train a small convolutional neural network, with two convolutional layers and a kernel of size 3, on MNIST. In this case, the GGN can be instantiated explicitly. We recall that the

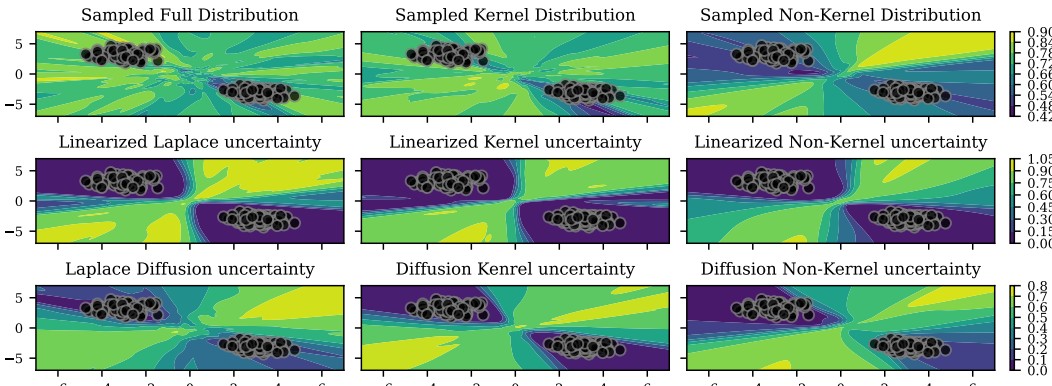

Figure 7: Decomposition of uncertainties of Laplace Approximation for the Gaussian mixture classification.

GGN is a sum of $\mathbf{J}_{\mathbf{w}'}(\mathbf{x}_n)^\top \mathbf{H}(\mathbf{x}_n)\mathbf{J}_{\mathbf{w}'}(\mathbf{x}_n)$ over the dataset where $\mathbf{x}_n$ are the individual data points. We recall that the rank of the GGN is bounded by $NO$ where $N$ is the number of data points and $O$ is the output dimensions. This suggests considering partial sums by subsampling the data points gives us a GGN with a lower rank. Equivalently, this is GGN with a higher dimensional kernel, and hence the usual covariance from Laplace's approximation $(\text{GGN} + \alpha I)^{-1}$ has a higher contribution from the kernel subspace.

We consider multiple such subsamples and plot the training accuracy for samples from Laplace's approximation against the kernel subspace dimension. Here we see a clear trend that the underfitting in sampled Laplace decreases as the rank of GGN increases, or the contribution from the kernel component decreases. This serves to further support our suggested hypothesis that the underfitting in sampled Laplace is caused by its kernel component and is hence deeply related to reparameterizations.

### E.3 Additional benchmarks and results for image classification

**Training details:** We use a standard LeNet for the MNIST and FashionMNIST experiments and a smaller version of ResNet (Lippe, 2022), with 272,378 parameters consisting of 3 times a group of 3 ResNet blocks. We use those instead of the standard ResNets due to constraints on the computational budget for the CIFAR-10 experiments. We train LeNet with Adam optimizer and a learning rate of $10^{-3}$. For the ReNet we use SGD with a learning rate of $0.1$ with momentum and weight decay.

**Hyperparameters:** We benchmark Laplace diffusion against SWAG, diagonal Laplace, last-layer Laplace, and MAP in addition to linearised Laplace and sampled Laplace.

For choosing the prior precision for diagonal Laplace and last-Layer Laplace for each benchmark we do a grid search over the set $\{0.1, 1.0, 5.0, 10.0, 50.0, 100.0\}$. For Laplace diffusion, sampled Laplace, and Linearised to ensure that the comparison can validate the theory it is preferable to have the same prior precision for all of these methods. So we only do the grid search to tune the prior precision for sampled Laplace and use this for all three methods. We keep the hyperparameters for these three methods as similar as possible to have the most informative comparisons.

For Laplace diffusion on MNIST and FMNIST, we simulate the diffusion with a step size of $0.05$, with 2000 Lanczos iterations and we predict using 20 MC samples. For sampled Laplace and Linearised Laplace, we also use 2000 Lanczos iterations and we predict using 20 MC samples. For the CIFAR-10 experiments, we simulate the diffusion with a step size of $0.2$, with 5000 Lanczos iterations and we predict using 20 MC samples. For sampled Laplace andlLinearised Laplace use the same number of Lanczos iterations and MC samples.

For SWAG we use a learning rate of $10^{-2}$ with momentum of $0.9$ and weight decay of $3e^{-4}$ and the low-rank covariance structure in all experiments. For the MNIST and FMNIST experiments we collect 20 models and for the CIFAR-10 experiments we collect 3 models to sample from the posterior.

Last-layer Laplace is the recommended method by Daxberger et al. (2021c) so it should approximate the best performance one can get using various possible configurations. For the CIFAR-10 experi-

ments, the last layer of ResNet is too large to instantiate the full GGN matrix. So we instead use the last 1000 parameters of the model to construct the covariance matrix of the posterior.

Diagonal Laplace requires high prior precision to ensure it does not severely underfit in-distribution (similar to sampled Laplace). It often becomes almost deterministic. So we exclude it from the CIFAR results. This has also been observed by Deng et al. (2022) and Ritter et al. (2018).

All additional information about the experimental setup can be found in the submitted code.

### E.3.1 In-distribution fit and calibration

We extend Table 1 to benchmark Laplace's diffusion against various other Bayesian methods. Here we see that despite using the neural network predictive it is competitive with the best-performing Bayesian methods whereas Sampled Laplace performs significantly worse.

**MNIST**

|  | Conf. ($\uparrow$) | NLL ($\downarrow$) | Acc. ($\uparrow$) | Brier ($\downarrow$) | ECE ($\downarrow$) | MCE ($\downarrow$) |
|---|---|---|---|---|---|---|
| Laplace's diffusion | 0.988±0.001 | 0.040±0.007 | 0.986±0.002 | 0.022±0.003 | 0.146±0.011 | 0.773±0.050 |
| Sampled Laplace | 0.593±0.002 | 3.669±0.157 | 0.157±0.031 | 1.162±0.044 | 0.436±0.027 | 0.984±0.001 |
| Linearised Laplace | 0.967±0.002 | 0.295±0.041 | 0.930±0.004 | 0.111±0.006 | 0.239±0.032 | 0.862±0.044 |
| SWAG | 0.993±0.001 | 0.032±0.001 | 0.989±0.002 | 0.019±0.001 | 0.187±0.024 | 0.901±0.041 |
| Last-Layer Laplace | 0.991±0.001 | 0.047±0.003 | 0.987±0.002 | 0.021±0.001 | 0.198±0.034 | 0.721±0.091 |
| Diagonal Laplace | 0.963±0.005 | 0.078±0.011 | 0.976±0.003 | 0.038±0.005 | 0.095±0.004 | 0.692±0.029 |
| MAP | 0.992±0.002 | 0.042±0.004 | 0.988±0.000 | 0.021±0.000 | 0.198±0.041 | 0.685±0.091 |

**FMNIST**

|  | Conf. ($\uparrow$) | NLL ($\downarrow$) | Acc. ($\uparrow$) | Brier ($\downarrow$) | ECE ($\downarrow$) | MCE ($\downarrow$) |
|---|---|---|---|---|---|---|
| Laplace's diffusion | 0.900±0.001 | 0.275±0.016 | 0.906±0.007 | 0.141±0.006 | 0.108±0.015 | 0.729±0.092 |
| Sampled Laplace | 0.618±0.021 | 4.507±0.160 | 0.098±0.010 | 1.295±0.014 | 0.518±0.013 | 0.986±0.001 |
| Linearised Laplace | 0.897±0.003 | 0.423±0.014 | 0.862±0.005 | 0.207±0.006 | 0.147±0.017 | 0.756±0.048 |
| SWAG | 0.925±0.002 | 0.259±0.004 | 0.911±0.006 | 0.135±0.004 | 0.152±0.008 | 0.752±0.067 |
| Last-Layer Laplace | 0.914±0.001 | 0.280±0.016 | 0.901±0.006 | 0.144±0.004 | 0.131±0.003 | 0.673±0.083 |
| Diagonal Laplace | 0.862±0.009 | 0.323±0.022 | 0.889±0.010 | 0.165±0.011 | 0.102±0.003 | 0.660±0.051 |
| MAP | 0.914±0.000 | 0.279±0.015 | 0.904±0.006 | 0.143±0.004 | 0.125±0.010 | 0.609±0.033 |

**CIFAR-10**

|  | Conf. ($\uparrow$) | NLL ($\downarrow$) | Acc. ($\uparrow$) | Brier ($\downarrow$) | ECE ($\downarrow$) | MCE ($\downarrow$) |
|---|---|---|---|---|---|---|
| Laplace's diffusion | 0.948±0.004 | 0.403±0.007 | 0.889±0.005 | 0.180±0.009 | 0.264±0.048 | 0.879±0.038 |
| Sampled Laplace | 0.843±0.004 | 0.997±0.222 | 0.717±0.049 | 0.422±0.081 | 0.221±0.047 | 0.804±0.080 |
| Linearised Laplace | 0.951±0.007 | 0.614±0.020 | 0.863±0.001 | 0.222±0.002 | 0.337±0.022 | 0.789±0.035 |
| SWAG | 0.942±0.003 | 0.393±0.004 | 0.884±0.002 | 0.176±0.001 | 0.234±0.014 | 0.912±0.016 |
| Last-Layer Laplace | 0.953±0.004 | 0.343±0.033 | 0.899±0.001 | 0.155±0.007 | 0.260±0.000 | 0.884±0.029 |
| MAP | 0.960±0.003 | 0.333±0.030 | 0.913±0.007 | 0.143±0.009 | 0.282±0.002 | 0.932±0.006 |

### E.3.2 Robustness to dataset shift

In these experiments (Fig. 8), to measure in-distribution fit and calibration, we report accuracy, negative log-likelihood (NLL), and expected calibration error (ECE)—all evaluated on the standard test sets. We measure the robustness of dataset shift of various baselines by plotting the negative log-likelihood and the expected calibration error against shift intensity. The desired behavior is good in-distribution fit, as close as possible to MAP, and stable calibration errors and NLL under distribution shifts. We see that the Laplace's diffusion is competitive against other Bayesian methods.

### E.3.3 Out-of-distribution detection

We extend 2 to benchmark Laplace's diffusion against various other Bayesian methods for Out-of-Distribution Detection. Once again, we observe that, despite using the neural network predictive,

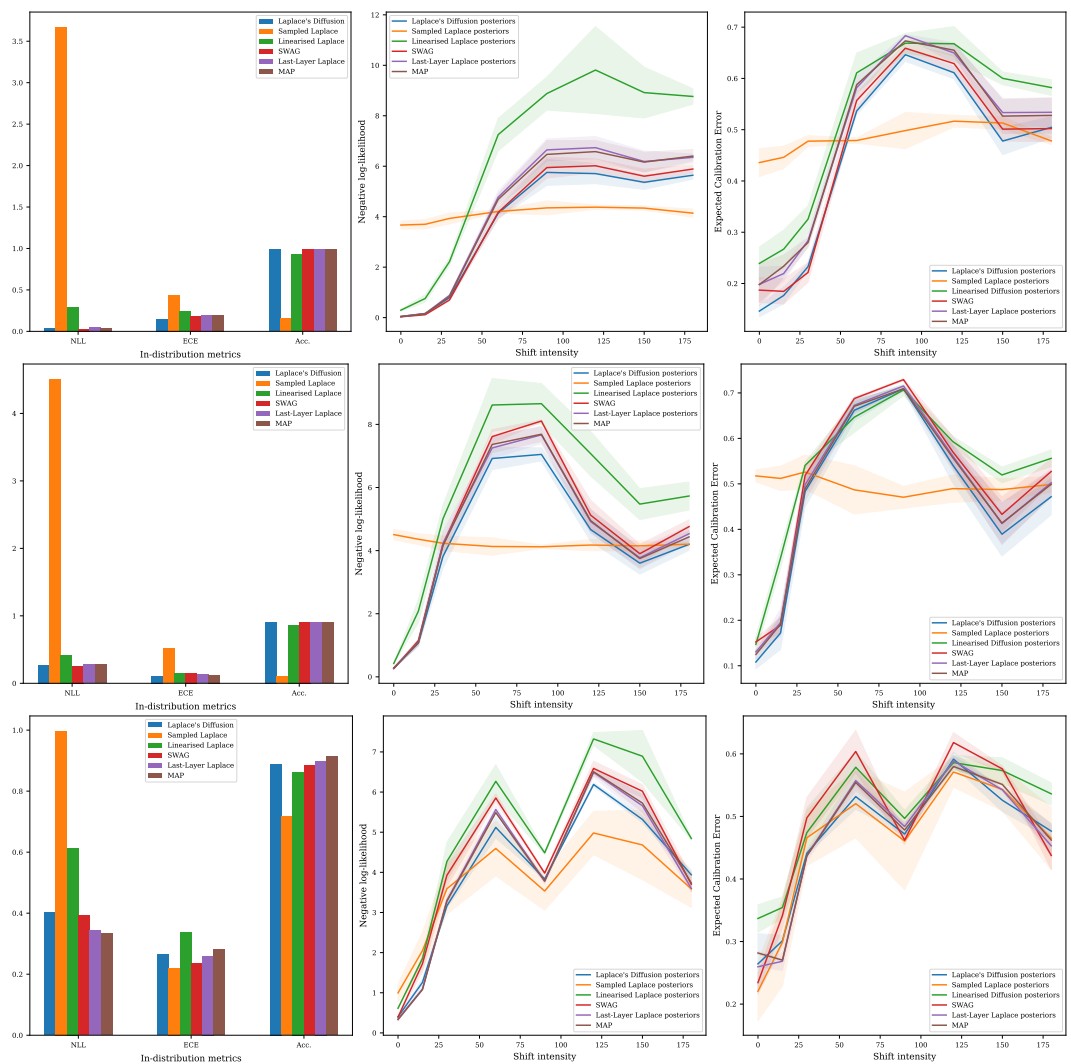

Figure 8: Model Fit and Calibration of various posterior sampling methods on in-distribution data(first column) and under distribution shift for MNIST(top row), Fashion MNIST(middle row) and CIFAR-10(bottom row). We use rotated MNIST, rotated FMNIST, and rotated CIFAR in the second and third columns. Shift intensities denote angles of rotation.

it is competitive with the best-performing Bayesian methods in terms of having a higher AUROC, whereas Sampled Laplace performs significantly worse.

**MNIST**

| Tested on | FMNIST | | EMNIST | | KMNIST | |
|---|---|---|---|---|---|---|
| | Conf.($\downarrow$) | AUROC($\uparrow$) | Conf.($\downarrow$) | AUROC($\uparrow$) | Conf.($\downarrow$) | AUROC($\uparrow$) |
| Laplace's diffusion | 0.810±0.031 | 0.911±0.037 | 0.947±0.007 | 0.629±0.021 | 0.817±0.015 | 0.930±0.009 |
| Sampled Laplace | 0.583±0.015 | 0.515±0.021 | 0.584±0.003 | 0.515±0.004 | 0.588±0.013 | 0.507±0.015 |
| Linearised Laplace | 0.895±0.027 | 0.715±0.086 | 0.934±0.004 | 0.536±0.028 | 0.863±0.009 | 0.757±0.018 |
| SWAG | 0.827±0.054 | 0.949±0.018 | 0.955±0.004 | 0.627±0.016 | 0.823±0.013 | 0.947±0.007 |
| Last-Layer Laplace | 0.842±0.046 | 0.904±0.033 | 0.958±0.005 | 0.625±0.028 | 0.837±0.009 | 0.935±0.004 |
| Diagonal Laplace | 0.770±0.026 | 0.866±0.030 | 0.907±0.005 | 0.617±0.009 | 0.747±0.013 | 0.897±0.005 |
| MAP | 0.850±0.035 | 0.907±0.038 | 0.959±0.005 | 0.634±0.019 | 0.837±0.010 | 0.938±0.006 |

**FMNIST**

| Tested on | MNIST | | EMNIST | | KMNIST | |
|---|---|---|---|---|---|---|
| | Conf.(↓) | AUROC(↑) | Conf.(↓) | AUROC(↑) | Conf.(↓) | AUROC(↑) |
| Laplace's diffusion | 0.734±0.039 | 0.759±0.045 | 0.730±0.020 | 0.741±0.010 | 0.746±0.012 | 0.749±0.023 |
| Sampled Laplace | 0.597±0.027 | 0.495±0.037 | 0.593±0.026 | 0.503±0.036 | 0.598±0.024 | 0.493±0.033 |
| Linearised Laplace | 0.817±0.022 | 0.625±0.050 | 0.813±0.003 | 0.628±0.013 | 0.816±0.014 | 0.624±0.020 |
| SWAG | 0.727±0.013 | 0.817±0.015 | 0.763±0.028 | 0.769±0.026 | 0.777±0.007 | 0.782±0.010 |
| Last-Layer Laplace | 0.757±0.019 | 0.761±0.031 | 0.760±0.017 | 0.735±0.018 | 0.772±0.008 | 0.747±0.021 |
| Diagonal Laplace | 0.652±0.033 | 0.767±0.032 | 0.682±0.019 | 0.719±0.033 | 0.696±0.025 | 0.728±0.032 |
| MAP | 0.759±0.019 | 0.757±0.032 | 0.762±0.018 | 0.730±0.015 | 0.773±0.010 | 0.743±0.024 |

**CIFAR-10**

| Tested on | CIFAR-100 | | SVHN | |
|---|---|---|---|---|
| | Conf.(↓) | AUROC(↑) | Conf.(↓) | AUROC(↑) |
| Laplace's diffusion | 0.790±0.002 | 0.851±0.002 | 0.764±0.008 | 0.862±0.010 |
| Sampled Laplace | 0.727±0.026 | 0.687±0.033 | 0.792±0.022 | 0.599±0.038 |
| Linearised Laplace | 0.818±0.005 | 0.837±0.006 | 0.809±0.033 | 0.854±0.024 |
| SWAG | 0.776±0.009 | 0.845±0.008 | 0.729±0.009 | 0.876±0.003 |
| Last-Layer Laplace | 0.789±0.006 | 0.864±0.001 | 0.786±0.029 | 0.868±0.015 |
| MAP | 0.797±0.004 | 0.873±0.001 | 0.792±0.034 | 0.878±0.017 |

### E.4 Short discussion on benchmarks

**Sparse Approximations of GGN.**   Our theoretical analysis is mainly concerned with the ideal versions of Laplace's approximations, where we consider the full GGN in the covariance without any approximations. However, it can also shed some light on other Bayesian methods.

It is common to use sparse approximations of the GGN when doing Laplace's approximations. Interestingly we observe that Laplace's approximations with sparse GGN such as diagonal Laplace, Last Layer, etc do not benefit from linearization to the same degree (Fig. 9).

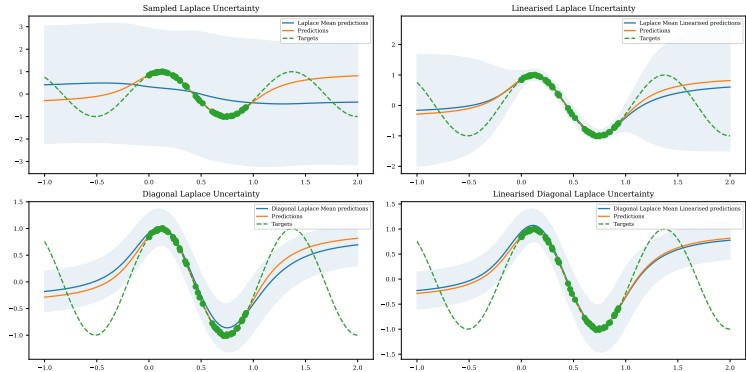

Figure 9: Predictive uncertainty of Laplace's approximation with neural network and linearized predictive (top row) and diagonal Laplace with neural network and linearized predictive (bottom row).

This is perfectly consistent with our analysis as we show that the benefit of linearization primarily comes from the Jacobian term in the linearized predictive and the GGN sharing a kernel. Diagonal and other sparse approximations do not share the spectral properties with the Jacobian in the linearized predictive. George et al. (2018) note the potential advantages of having the spectrum of the approximate curvature more aligned with the true curvature. This suggests future directions to improve various approximations to the GGN by accounting for reparameterizations.

**SWAG.** Another baseline that can be explained using our method is the SWAG. It has been shown that in (Li et al., 2021) SGD steps close to the optimum can be decomposed into a normal space component and a tangent space component. In our terminology, this can be thought of as a diffusion step in the Kernel manifold and a diffusion step in the Non-kernel manifold. Hence it can be shown that SWAG roughly approximates a diffusion-based posterior. Hence our analysis can provide some theoretical grounding for heuristic methods like SWAG.

