# OpenReview forum: "Reparameterization invariance in approximate Bayesian inference"
_NeurIPS.cc/2024/Conference — NeurIPS 2024 spotlight_

### Official Review · Reviewer_v5yb · 2024-07-03

**Soundness:** 4
**Presentation:** 3
**Contribution:** 4
**Rating:** 8
**Confidence:** 4

**Summary:**

The paper studies the effects of the invariance of Bayesian neural networks under reparametrization on their approximate posterior inference. The primary effect is the ambiguity on the uncertainty of the inferred function as it can be represented by multiple reparameterizations, each of which may be assigned a different uncertainty estimate. The paper takes Laplace approximation as a case and studies its properties when used for Bayesian neural net inference using linear algebraic and differential geometric tools. The paper derives an algorithm from its key theoretical outcomes and shows it to outperform its counterparts in various downstream uncertainty quantification tasks.

**Strengths:**

* This is a very high-calibre paper, a complete piece of work, that studies an original and significant problem with impressive theoretical rigor.

* The presentation is extremely clear. Figures 1 and 2 are truly helpful to the reader to grasp the visual intuition behind the studied problem and the proposed solution.

* The compilation of theoretical tools such as the a parameter manifold on the Generalized Gauss-Newton (GGN) approximation and a diffusion defined on it are original, advanced, and elegant.

* The reported numerical results are both comprehensive and strongly in favor of the central claims.

**Weaknesses:**

* Overall this is excellent piece of work with a mature write-up. Just its presentation may be improved tiny bit. For instance, certain terms are used very early on in the paper such as the introduction and Figure 1 in their meanings a bit unusual for the broad ML research readership, such as a "kernel" and "diffusion". While their allocation in the particular context is appropriate, reading experience may be improved if their meanings are made more explicit, at least verbally, in their first use.

**Questions:**

* How does the argumentation in the first paragraph of Section 3 come together with the main message of the paper? Is the problem not instead that the posterior assigns separate masses to different weights that represent the same functional form? How exactly does this produce a pathology in the posterior besides the need to detect multiple equivalent representations and add their masses? Why do we actually need to design the related degrees of freedom? Overall I buy the argument and see the problem, but my point is that probably this bridge paragraph may be misleading the reader.

 * What does f_lin^w precisely mean in Eq 4? Is it the first-order Taylor approximation of f?

**Limitations:**

The paper briefly discusses its limitations in the final paragraph of the Conclusion section. The potential negative societal impacts of the studied work are not discussed probably because they are thought to be inapplicable.

---

> ### Author Rebuttal · Authors · 2024-08-07
>
> We thank the reviewer for the excellent review and we are very grateful for a thorough engagement with our work and for pinpointing its highlights.
>
> We acknowledge your point (under Weaknesses) about providing a bit more explanation of unconventional terms such as kernel (that in machine learning often means something entirely different). We will take this feedback to heart and update the manuscript accordingly.
>
> Regarding the opening paragraph of Sec. 3, then we acknowledge that, in particular, the phrasing “pathological representations” is too vague and opens the door to misinterpretations. We suggest to rephrase the sentence
>  “However, as we have argued, traditional approximate Bayesian inference does not correctly capture this, leading to pathological representations”
> into
> “However, as we have argued, traditional approximate Bayesian inference does not correctly capture this and assigns different probability measures to identical functions.”
> We believe such a change clarifies the text, but are happy to take further feedback.
>
> Yes, $f_{lin}^w$ is exactly the mentioned Taylor expansion. The equation is given in line 107 of the submitted manuscript, but it would have been better to already introduce this notation around line 64. We will make this change.
>
> We hope this answers your questions and concerns, and are happy to discuss further. Once again, we are grateful for the favorable review that exactly captures our intention, and we hope that you will continue to fight for the paper.

---

> > ### Comment · Reviewer_v5yb · 2024-08-08
> > **Keep score**
> >
> > Thanks for the clarification. Rephrasing the sentence in the suggested way will indeed do the trick. While I do agree with the other reviewers that the method has an computational overhead and experiments may be improved, I still think that the technical contribution is solid and valuable and the other issues are rather secondary. Hence, I tend to keep my score for now.

---

### Official Review · Reviewer_gWds · 2024-07-11

**Soundness:** 3
**Presentation:** 3
**Contribution:** 3
**Rating:** 5
**Confidence:** 3

**Summary:**

The paper addresses the problem of maintaining invariance under reparameterization of Bayesian Neural Networks (BNNs). This issue undermines the reliability of BNNs since different parameterizations of identical functions produce different posterior densities. Through theoretical analysis and empirical validation, the authors demonstrate the effectiveness of their approach by extending a geometric view (in Linearized Laplace) and using a Riemannian diffusion process to achieve reparameterization invariance.

**Strengths:**

* Proposed Riemannian diffusion process incorporates the properties of the linearized Laplace approximation into the original neural network prediction, resulting in a better posterior fit.
* The method leverages advanced mathematical tools from differential geometry, offering a sophisticated approach to maintaining reparameterization invariance.
* In both in-distribution and out-of-distribution scenarios, the proposed method consistently outperforms other approaches.
* The paper is well-organized and well-written, with each section building logically on its predecessor.

**Weaknesses:**

* Although the proposed method is theoretically sound, its computational complexity is higher than simpler approximations. This could limit its practicality for very large-scale neural networks.
* Similar to previous works in Linearized Laplace, the need for new computational pipelines to handle the GGN matrix and its induced structure is acknowledged but not fully addressed in the paper.
* Although the experiments are comprehensive, they demonstrate results on standard datasets and relatively small to medium-sized neural networks.Testing the method on more diverse datasets and larger-scale models would strengthen the empirical validation.

**Questions:**

* Although mentioned in related work, I feel that this paper needs comparative experiments with methods such as Riemannian Laplace and (Bergman et al., 2024) and Connectivity Laplace (Kim et al., 2023).
* As mentioned in Weakness section, I would recommend the author to evaluate the Riemannian Diffusion in other domain dataset & architectures (e.g., Transformers).

**Limitations:**

Limitations are mentioned in "Weakness" and "Questions".

---

> ### Author Rebuttal · Authors · 2024-08-07
>
> We thank the reviewer for their valuable review.
>
> - “Although the proposed method is theoretically sound, its computational complexity is higher than simpler approximations. [...]”
>
> While this is a valid comment we believe it stems from a slight misunderstanding of the main point that is being made with our experiments.
>
> The main goal of this paper is to understand the uncertainties obtained from Laplace approximations and more generally to understand some of the problems encountered in Bayesian deep learning. More specifically decomposing the uncertainty into its two fundamental components and how they interact with the choice of the predictive (neural network or the linearized neural network). Our purpose is not to advance the state-of-the-art of Bayesian methods.
>
> One way to think about the experiments is as follows:
> Observation: Linearized Laplace underfits less (in-distribution) than Sampled Laplace.
> Hypothesis: This difference is due to improper accounting of the reparameterization properties in the Sampled Laplace.
> Theory: The theory presented in the paper provides mathematical justification for this hypothesis.
> Experiment: We build a method that is as close to Sampled Laplace as possible while also accounting for the reparameterization properties of the neural network. This is the Laplace diffusion which turns out to not underfit in-distribution, hence supporting the extensive theory developed in the paper.
>
> For this reason, we voluntarily spend the extra compute to get as close as possible to the theoretical ideal. Specifically, for a $P$ parameters network, computing $S$ samples by approximating the diffusion with $T$ discrete time steps and a low-rank $K$ approximation of the GGN requires $O(KP + SP)$ in memory and $O(K^2PST)$ in time, i.e. it is $ST$ times as expensive as linearized Laplace.
>
> We could have made the method more scalable by fixing low values for $K$ or $T$. However, the resulting experiments can no longer validate the theory due to the brutal approximations. This is also why we only include comparison with baselines in the appendix omitting them from the main paper, where we only compare the different predictive functions.
>
> Of course, we do not disregard the importance of building practical, performant, and scalable Bayesian methods. In future work, we intend to build on the here-presented theoretical insights to achieve this exact goal. However, understanding why Bayesian deep learning struggles (this paper) is a prerequisite. The main takeaway of this paper is not a particular method but a general principle: Think about reparameterization issues when building Bayesian methods.
>
> - “Similar to previous works in Linearized Laplace, the need for new computational pipelines to handle the GGN matrix and its induced structure is acknowledged but not fully addressed in the paper.”
>
> Indeed, our paper concludes with a brief discussion of this computational challenge. We intended to acknowledge the inherent difficulty and to draw more attention to this problem. That being said, the paper does contribute advances in this direction as our implementation uses efficient matrix-free access to the GGN using automatic differentiation combined with iterative solvers to obtain samples from the Laplace approximation. This is a significant improvement over existing implementations. We deemphasize these computational aspects of our work and only discuss them implicitly because our main focus is on the theoretical explanation.
>
> - “Although the experiments are comprehensive, they demonstrate results on standard datasets and relatively small to medium-sized neural networks. Testing the method on more diverse datasets and larger-scale models would strengthen the empirical validation.”
>
> We have chosen to replicate the key experiments from the Laplace Redux paper [1] which is an important reference for Laplace approximations. Their largest experiments are also on CIFAR-10 with ResNets so we don’t believe the scale of our experiments is too unreasonable. Furthermore, scaling to larger datasets and models necessarily requires making some fairly brutal approximations, by doing subnetwork inference, KFAC, diagonal approximations, etc. We are not at all opposed to this but hopefully, it is clear from the previous response why such approximations are inappropriate in the context of theory-validation.
>
> - “Although mentioned in related work, I feel that this paper needs comparative experiments with methods such as Riemannian Laplace and (Bergman et al., 2024) and Connectivity Laplace (Kim et al., 2023).”
>
> As mentioned above, our main experimental goal is to validate the theory. Further note that the public code for the first reference is orders of magnitude slower than our code and it is infeasible to run on as large models as we do. The public repository for the second reference has no documentation and no code for reproducing the experiments in their paper. Consequently, it would be highly non-trivial to run them on our benchmarks.
>
> - “As mentioned in Weakness section, I would recommend the author to evaluate the Riemannian Diffusion in other domain dataset & architectures (e.g., Transformers).”
>
> To our knowledge, it's not standard practice to evaluate Laplace approximations on transformers. We are only aware of [2] which evaluates Laplace approximations on transformers and they perform Laplace approximations only on the low-rank adaptation weights. Hopefully from the previous response, it is clear why such approximations are not appropriate for our experiments given our purpose.
>
> We thank the reviewer for their efforts and we urge them to reconsider their assessment of the experiments in light of the additional context we have provided above.
>
> [1] Daxberger, Erik, et al. "Laplace redux-effortless Bayesian deep learning." Advances in Neural Information Processing Systems 34 (2021)
>
> [2] Yang, Adam X., et al. "Bayesian low-rank adaptation for large language models." arXiv:2308.13111 (2023)

---

> ### Comment · Reviewer_gWds · 2024-08-13
>
> Thanks for the detailed response from the authors.
>
> The authors claim I misunderstood the paper, but I don't believe that's true. Linearized Laplace outperforms (sampled) Laplace, and there are already other theoretical approaches to the problem. (See Theorem 4.1 in [1])
>
> In addition, since the experiment I requested is applied to a very simple transformer, I don't think it is impossible.
>
> As a result, I will maintain the current score.
>
>
>
> [1] Kim, SungYub, Kyungsu Kim, and Eunho Yang. "GEX: A flexible method for approximating influence via Geometric Ensemble." Advances in Neural Information Processing Systems 36 (2023).

---

> > ### Author Response · Authors · 2024-08-13
> >
> > > The authors claim I misunderstood the paper, but I don't believe that's true. Linearized Laplace outperforms (sampled) Laplace, and there are already other theoretical approaches to the problem. (See Theorem 4.1 in [1])
> >
> > This paper (already referenced by us) does not compare Linearized and Sampled Laplace. Further, note that they make no theoretical links between linearization and reparametrization invariance and they study the Hessian instead of the GGN.
> >
> > > In addition, since the experiment I requested is applied to a very simple transformer, I don't think it is impossible.
> >
> > It is unclear to us what the exact experiment you suggest. Previously we interpreted your review to mean a standard transformer in an NLP task. We reemphasize that this is not a usual benchmark for evaluating Bayesian methods especially Laplace Approximations where standard regression and image classification tasks are far more popular. Hence it is not obvious what priors are appropriate, what dataset should be used, and what metrics should be reported. These are important questions and we do not aim to settle them here; Tristan et al. discuss some of the pitfalls and difficulties of such experiments.
> >
> > If the reviewer means a Vision Transformer (smaller version with only a few million parameters) on the standard image classification task, this is very much within reach. In fact, we even have code to run such experiments. However, doing so would require us to make some large approximations (limited Lanczos iterations and diffusion steps). The resulting experiment would, thus, not serve for theory validation, but we are happy to add them nonetheless if you see value therein.
> >
> > Cinquin, Tristan, et al. "Pathologies in priors and inference for Bayesian transformers." arXiv preprint arXiv:2110.04020 (2021). (edited)

---

### Official Review · Reviewer_12ke · 2024-07-14

**Soundness:** 3
**Presentation:** 3
**Contribution:** 3
**Rating:** 7
**Confidence:** 3

**Summary:**

The authors provide theoretical analysis of the underfitting problem of Laplace approximation. Specifically, they show that the underfitting of Laplace approximation is due to the approximate posterior covariance is not invariant under reparameterization. Moreover, they propose a reparametrization invariant diffusion posterior to address undefitting. The method are evaluated over standard image classification benchmarks. The quality of the uncertainty for both in-distribution and out-of-distribution tasks are considered.

**Strengths:**

1. The paper provides, to my knowledge, the first theoretical justification of why Laplace approximation sufferes from underfitting: the approximate posterior covariance is not reparameterization invariance.
2. Given the theoretical analysis, the idea of reparametrization invariant posterior is natural and proves to be effective empirically.
3. The paper is easy to follow and well organized.

**Weaknesses:**

1. The proposed method, as mentioned by the authors as well, sufferes from more expensive computation. It would be good to report the time metric for readers to properly assess the practicability of the method.
2. The datasets considered are a bit outdated and the networks considred seem to be quite small, such that overall the performance is on the lower end (e.g. <90% acc. for CIFAR10). Recent BDL papers typically consider larger datasets (such as ImageNet) and deeper networks (e.g. [1]). Furthermore, last-layer Laplace seems to be competitive with or even outperform the proposed method in some experiments.
3. The method is tailored to improving Laplace approximation, and there is no discussion related to any other approximate inference techiques (e.g. variational inference).

References
[1] Antoran et al. Sampling-based inference for large linear models with application to linearised Laplace. ICLR 2023.

**Questions:**

See weakness.

**Limitations:**

The authors do not address potential negative social impact since the paper is predominantly theoretical.

---

> ### Author Rebuttal · Authors · 2024-08-07
>
> Thanks for the excellent review, which we will reply to in parts.
>
> - “The paper provides, to my knowledge, the first theoretical justification of why Laplace approximation sufferes from underfitting: the approximate posterior covariance is not reparameterization invariance.”
>
> This is correct. Additionally, we are the first to provide a complete characterization of continuous reparameterizations of a neural network. Certain reparameterizations such as scale reparameterizations for ReLUs are already known but we provide a complete characterization of these in terms of the kernel manifold. Furthermore, to the best of our knowledge, we are the first to consider data-dependent reparameterizations rather than only global reparameterization (a subset of the former).
>
> - “The proposed method, as mentioned by the authors as well, suffers from more expensive computation. It would be good to report the time metric for readers to properly assess the practicability of the method.”
>
> For a $P$ parameters network, computing $S$ samples by approximating the diffusion with $T$ discrete time steps and a low-rank $K$ approximation of the GGN requires $O(KP + SP)$ in memory and $O(K^2PST)$ in time, i.e. it is $ST$ times as expensive as linearized Laplace.
>
> However, we want to emphasize that our main goal here is to understand the uncertainties obtained from Laplace approximations and more generally to understand some of the problems encountered in Bayesian deep learning. More specifically decomposing the uncertainty into its two fundamental components and how they interact with the choice of the predictive (neural network or the linearized neural network). Our purpose is not to advance the state-of-the-art of Bayesian methods.
>
> One way to think about the experiments is as follows:
>
> Observation: Linearized Laplace underfits less (in-distribution) than Sampled Laplace.
> Hypothesis: This difference is due to improper accounting of the reparameterization properties in the Sampled Laplace.
> Theory: The theory presented in the paper provides mathematical justification for this hypothesis.
> Experiment: In addition to the theoretical justification, we build a method that is as close to Sampled Laplace as possible while also accounting for the reparameterization properties of the neural network. This is the Laplace diffusion which turns out to not underfit in-distribution, hence supporting the extensive theory developed in the paper.
>
> For this reason, we voluntarily pay the extra computational factor $T$ to get as close as possible to the theoretical ideal.
> For example, we could have made the method more scalable by fixing low values for $K$ or $T$. While such approximations might be more practical and scalable, the resulting experiments can no longer validate the theory due to the brutal approximations. This is also why we only include comparison with baselines in the appendix omitting them from the main paper, where we only compare the different predictive functions.
>
> Of course, we do not disregard the importance of building practical, performant, and scalable Bayesian methods. In future work, we intend to build on the here-presented theoretical insights to achieve this exact goal. However, understanding why Bayesian deep learning struggles (this paper) is a prerequisite. The main takeaway of this paper is not a particular method but a general principle: Think about reparameterization issues when building Bayesian methods.
>
> Further discussion of these issues and time complexity is available in Appendix E.1.
>
> - “The datasets considered are a bit outdated and the networks considred seem to be quite small [...]”
>
> For the experiments, we have chosen to replicate the key experiments from the Laplace Redux paper [1] which is an important reference for Laplace approximations. Their largest experiments are also on CIFAR-10 with ResNets so we don’t believe the scale of our experiments is too unreasonable.
>
> Secondly, we are certainly aware of the recent works that do Laplace approximation on larger models (see references in the paper). But they all invariably rely on fairly brutal approximations such as KFAC, last-layer, or LoRa. We are not at all opposed to this but hopefully, it is clear from the previous response why such approximations are inappropriate in the context of our experiments: these aim for theory-validation rather than the usual focus on performance and scalability.
>
> - “The method is tailored to improving Laplace approximation, and there is no discussion related to any other approximate inference techiques (e.g. variational inference).”
>
> While it’s true that our primary focus is on Laplace approximations, we believe that the tools developed here are more generally applicable. The theory of reparameterizations can find applicability in various topics such as continual learning and in studying the loss landscape. The issues considered here are also relevant to other Bayesian methods. It's not possible to explicate this fully but we have a brief discussion of this direction in appendix E.4. Another possible direction is Variational Inference with the diffusion posterior as a variational family. Secondly, we believe that the underfitting of Sampled Laplace is an important open problem in the community, which should not be understated. For example, this has also been discussed in a recent position paper [2]
>
> We thank the reviewer for their efforts and we urge them to reconsider their assessment of the experiments in light of the additional context we have provided above.
>
> [1] Daxberger, Erik, et al. "Laplace redux-effortless Bayesian deep learning." Advances in Neural Information Processing Systems 34 (2021): 20089-20103.
>
> [2] Papamarkou, Theodore, et al. "Position paper: Bayesian deep learning in the age of large-scale AI." arXiv preprint arXiv:2402.00809 (2024)

---

> > ### Comment · Reviewer_12ke · 2024-08-12
> >
> > Thank you for your detailed rebuttal, which addresses most of my concerns, and I decide to raise my score to 7.

---

### Official Review · Reviewer_AZGF · 2024-07-14

**Soundness:** 3
**Presentation:** 2
**Contribution:** 3
**Rating:** 5
**Confidence:** 3

**Summary:**

This work questions why the linearized Laplace approximation can be effective for uncertainty estimation, whereas the vanilla Laplace approximation can lead to poor performance, such as underfitting. This work argues that this degradation is attributed to the fact that the random function derived from each parameterization of a DNN may not form an invariant function space.

Specifically, since the null space $ \text{ker} (\text{GGN}_{w}) $ is not empty for over-parameterized NNs, sampling the random function containing elements of null space could result in an inconsistent random function. This can lead to under-fitting issues as observed in Laplace approximation.

Therefore, the authors believe that sampling the random function on $ \text{im} (\text{GGN}_{w}) $, i.e., the linearized Laplace, could be more effective than the Laplace method.

Furthermore, for a given weight $w$, they explore how to sample the same function with  $ f(w) $, i.e., $ f(g(w)) = f(w) $, by sampling the weight on a specific manifold $ g(w) $. To this end, they employ the concept of quotient space to elaborately define the manifold and diffusion to sample $ w $ from the manifold $ g(w) $.

**Strengths:**

* This work reasonably reveals why the linearized Laplace approximation can be more effective than the standard Laplace approximation. This explanation appears to be novel.

* This work attempts to justify the above reasoning using a mathematical elaboration framework.

**Weaknesses:**

* The lack of explanation of background knowledge, such as quotient space and Riemannian manifold, makes the paper difficult to understand.

* Although it appears effective compared to Laplace and Linearized Laplace, the performance improvement seems marginal when compared to other baselines such as SWAG and Last-layer Laplace.

**Questions:**

* To confirm my understanding, does the Laplace diffusion denote that for a given $w$, the sample functions $ { w_t }$ are obtained by the update rule using $\text{GGN}^{+}$ as described in the SDE on the manifold?

* Does $\text{GGN}^{+}$ mean the positive definite matrix of $\text{GGN}$, which is obtained by applying SVD on $\text{GGN}$ and then using the eigenvectors with positive eigenvalues?

* Is the Laplace diffusion the post-hoc method, meaning that $w$ is first obtained by MAP inference, i.e., $w_{\text{MAP}} = R^{f}_{\mathcal{X}}(w)$, and then $w_t$ is obtained by Laplace diffusion with $\text{GGN}^{+}$?

* Compared to the performance of SWAG described in attached appendix, the performance of the Laplace diffusion does not seem significantly improved.
In this context, for a given $w$, is it important to sample the weight parameter according to the invariant Riemannian manifold?

* Rather, as considering that SWAG focuses on finding good neighborhood of $w$ in training procedure and obtains comparable performance, isn't it more important to focus on how to find good $w$ (for example, $ w_{\text{swa}} $ ) and explore the subspace of $w$? I am just curious about your opinion on this.

**Limitations:**

See above weaknesses and limitations.

---

> ### Author Rebuttal · Authors · 2024-08-07
>
> We thank the reviewer for their valuable review and questions, which will further improve the paper.
>
> - “The lack of explanation of background knowledge, such as quotient space and Riemannian manifold, makes the paper difficult to understand.”
>
> We acknowledge this issue, which is due to the space constraints of a conference paper. Because of the additional allowed page, we will be happy to extend the background section to cover the necessary knowledge in Riemannian geometry to benefit future readers.
>
> - “Although it appears effective compared to Laplace and Linearized Laplace, the performance improvement seems marginal when compared to other baselines such as SWAG and Last-layer Laplace.”
>
> While this is a valid comment we believe it stems from a slight misunderstanding of the main point of our experiments.
>
> The main goal of this paper is to understand the uncertainties obtained from Laplace approximations and more generally to shed light on some of the problems encountered in Bayesian deep learning. More specifically decomposing the uncertainty into its two fundamental components and how they interact with the choice of the predictive (neural network or the linearized neural network). Our purpose is not to advance the state-of-the-art of Bayesian methods.
>
> We include the baselines in the appendix and omit them from the main paper for precisely this reason. We want to validate the theory in realistic settings while staying as close as possible to the theoretical ideal (these choices are also discussed in Sec. 7 and Appendix E.1).
>
> One way to think about the experiments is as follows:
> Observation: Linearized Laplace underfits less (in-distribution) than Sampled Laplace.
> Hypothesis: This difference is due to improper accounting of the reparameterization properties in the Sampled Laplace.
> Theory: The theory presented in the paper provides mathematical justification for this hypothesis.
> Experiment: In addition to the theoretical justification, we build a method that is as close to Sampled Laplace as possible while also accounting for the reparameterization properties of the neural network. This is the Laplace diffusion which turns out to not underfit in-distribution, hence supporting the extensive theory developed in the paper.
>
> Of course, we do not disregard the importance of building practical, performant, and scalable Bayesian methods. In future work, we intend to build on the here-presented theoretical insights to achieve this exact goal. However, understanding why Bayesian deep learning struggles (this paper) is a prerequisite. The main takeaway of this paper is not a particular method but a general principle: Think about reparameterization issues when building Bayesian methods.
>
> To confirm my understanding, does the Laplace diffusion denote that for a given 𝑤, the sample functions 𝑤𝑡 are obtained by the update rule using GGN+ as described in the SDE on the manifold?
> Diffusion can be performed by simulating the SDE in the kernel manifold and the non-kernel manifold or both (using alternating steps, for example). In Figs. 2 and 7 we perform all 3 diffusions. For the larger experiments (Sec. 7), your description is correct. We only perform the diffusion in the non-kernel manifold. The reasons for this are discussed in Section 5.
>
> - "Does GGN+ mean the positive definite matrix of GGN, which is obtained by applying SVD on GGN [...]?"
>
> Essentially yes. However, we use a Lanczos decomposition of the GGN instead of SVD.
>
> Is the Laplace diffusion the post-hoc method, meaning that 𝑤 is first obtained by MAP inference, i.e., 𝑤 MAP=𝑅𝑋𝑓(𝑤), and then 𝑤𝑡 is obtained by Laplace diffusion with GGN+?
>
> Yes, it is a post-hoc method. As stated above we want to build a method as close to Sampled Laplace as possible which additionally is reparameterization invariant. Since Sampled Laplace is typically a post-hoc method it makes sense that Laplace diffusion should be as well.
>
> - "Compared to the performance of SWAG described in attached appendix, the performance of the Laplace diffusion does not seem significantly improved. In this context, for a given 𝑤, is it important to sample the weight parameter according to the invariant Riemannian manifold? Rather, as considering that SWAG focuses on finding good neighborhood of 𝑤 in training procedure and obtains comparable performance, isn't it more important to focus on how to find good 𝑤(for example, 𝑤 swa) and explore the subspace of 𝑤? I am just curious about your opinion on this."
>
> This is an interesting point of discussion, which we will approach in steps.
> First, note that while the mode (i.e. the MAP) of the Gaussian approximate posterior is very important, the reparametrization issue we discuss is mainly related to the covariance. SWAG restricts the covariance to a subspace.
> SWAG involves several heuristics that are hard to characterize theoretically, so here we consider a simplified version:
> Fix a mode parameter $w_m$, then perform one SGD step to find a new parameter $w_1$. Repeat this (starting from $w_m$) several times.
> Compute the empirical covariance of $w_1,w_2,..$ and use this for the Gaussian posterior approximation.
> It’s important to clarify that the kernel directions (i.e. 0-eigenvalue directions of the GGN) depend on the dataset. We can, thus, consider both the full-dataset-kernel directions and the batch-kernel directions. Notably, the full-dataset-kernel is always contained in the batch-kernel.
> The covariance obtained by simplified-SWAG is then guaranteed to be 0 along the full-dataset-kernel direction.
> This shows that the simplified-SWAG approximate posterior is locally reparametrization invariant, thereby demonstrating how the developed theory can help us understand existing methods and eventually develop new ones.
>
> We thank the reviewer for their efforts and we encourage them to reconsider their assessment of the experiments in light of the additional context we have provided above.

---

> > ### Comment · Reviewer_AZGF · 2024-08-13
> >
> > Thank you for your responses to my questions. During the rebuttal period, I was able to resolve my concerns and gain a deeper understanding of the importance of the reparametrization invariance that this work aims to highlight. I am inclined to accept this work and will therefore maintain my score.

---

> ### Author Response · Authors · 2024-08-13
>
> > During the rebuttal period, I was able to resolve my concerns and gain a deeper understanding of the importance of the reparametrization invariance that this work aims to highlight.
>
> Thanks for the support. We very much agree that reparametrization invariance is important. When models grow in size (e.g. increasing depth), we spend a greater proportion of the parameters on reparametrizations. Ten years ago, models were sufficiently small that reparametrizations could practically be ignored, but today they are causing sufficiently many problems that we are forced to better understand them. We believe that reparametrizations are the reason why Bayesian approximations do not work well with contemporary models, even if they work for small models. This is why it's essential that we form an understanding of reparametrizations.

---

### Author Rebuttal · Authors · 2024-08-07

We are grateful to the four reviewers who all argue in favor of acceptance.

We observe a general agreement that the developed theory is both novel and sheds significant light on the difficulties of Bayesian deep learning induced by reparametrizations. Some concerns have been raised about the experimental part of the paper. Specifically, there were concerns about the scalability and performance of Laplace diffusion. Here we want to clarify that the stated intent of the experiments is to support the developed theory rather than to develop a state-of-the-art and scalable method (despite Laplace diffusion’s favorable performance). For this reason, we do not optimize the method for performance but keep it as close as possible to Sampled Laplace and we also avoid approximations that might make it more scalable. We argue that the work presented provides the theoretical foundation opening up new insights that will allow for the development of new and more efficient methods that will be both computationally efficient and accurate. The main takeaway of this paper is not a particular method but a general principle: “ Think about reparameterization issues when building Bayesian methods”

We reply to the individual reviews below.

---

### Decision · Program_Chairs · 2024-09-25

**Decision:**

Accept (spotlight)

**Comment:**

This work studies the reparameterization invariance of Laplace approximations in Bayesian neural networks. The study begins by noting that the linearized Laplace suffers less from underfitting than the sample Laplace approach. This observation leads to the hypothesis that the discrepancy arises from a failure to properly account for reparameterization invariance in the nonlinear setting. The authors provide a rigorous theoretical analysis to support this claim and propose a modification to the sample Laplace method to address the issue. The empirical results strongly support the theoretical findings. All reviewers agree that this work presents an original perspective, addresses a significant problem, and offers a complete theory backed by convincing empirical evidence. Therefore, I recommend acceptance.

The authors are encouraged to include a detailed introduction to background concepts, such as quotient spaces and Riemannian manifolds, in the final version.